# Room-temperature crystallography reveals altered binding of small-molecule fragments to PTP1B

Tamar Skaist Mehlman[1,2], Justin T Biel[3], Syeda Maryam Azeem[1†], Elliot R Nelson[4], Sakib Hossain[1], Louise Dunnett[4,5], Neil G Paterson[4], Alice Douangamath[4,5], Romain Talon[4], Danny Axford[4], Helen Orins[1], Frank von Delft[4,5,6,7], Daniel A Keedy[1,8,9]*

[1]Structural Biology Initiative, CUNY Advanced Science Research Center, New York, United States; [2]PhD Program in Biochemistry, CUNY Graduate Center, New York, United States; [3]Department of Bioengineering and Therapeutic Sciences, University of California, San Francisco, San Francisco, United States; [4]Diamond Light Source, Didcot, United Kingdom; [5]Research Complex at Harwell, Harwell Science and Innovation Campus, Didcot, United Kingdom; [6]Centre for Medicines Discovery, Nuffield Department of Medicine, University of Oxford, Oxford, United Kingdom; [7]Department of Biochemistry, University of Johannesburg, Johannesburg, South Africa; [8]Department of Chemistry and Biochemistry, City College of New York, New York, United States; [9]PhD Programs in Biochemistry, Biology, and Chemistry, CUNY Graduate Center, New York, United States

*For correspondence: dkeedy@gc.cuny.edu

Present address: †PhD Program in Biology, CUNY Graduate Center, New York, United States

**Abstract** Much of our current understanding of how small-molecule ligands interact with proteins stems from X-ray crystal structures determined at cryogenic (cryo) temperature. For proteins alone, room-temperature (RT) crystallography can reveal previously hidden, biologically relevant alternate conformations. However, less is understood about how RT crystallography may impact the conformational landscapes of protein-ligand complexes. Previously, we showed that small-molecule fragments cluster in putative allosteric sites using a cryo crystallographic screen of the therapeutic target PTP1B (Keedy et al., 2018). Here, we have performed two RT crystallographic screens of PTP1B using many of the same fragments, representing the largest RT crystallographic screens of a diverse library of ligands to date, and enabling a direct interrogation of the effect of data collection temperature on protein-ligand interactions. We show that at RT, fewer ligands bind, and often more weakly – but with a variety of temperature-dependent differences, including unique binding poses, changes in solvation, new binding sites, and distinct protein allosteric conformational responses. Overall, this work suggests that the vast body of existing cryo-temperature protein-ligand structures may provide an incomplete picture, and highlights the potential of RT crystallography to help complete this picture by revealing distinct conformational modes of protein-ligand systems. Our results may inspire future use of RT crystallography to interrogate the roles of protein-ligand conformational ensembles in biological function.

## Editor's evaluation

Based on two room-temperature X-ray crystallographic screens of the phosphatase PTP1B against two sets of chemical fragments, and by comparing the results from a previous cryo screen, the authors report the important observation that, in addition to overlapping but non-identical sets of hits compared to the cryo screen, the room-temperature screens lead to significant differences in

terms of binding sites and poses for some of the hits. The study provides compelling support for the use of room-temperature X-ray crystallography in early-stage drug discovery and highlights that temperature should be used as a parameter in efforts to extract additional insight from such analyses.

## Introduction

Of the ~150,000 protein crystal structures in the public Protein Data Bank (PDB) (*Berman et al., 2000*), ~122,000 (~81%) have a non-polymer ligand modeled, and many thousands more reside in private pharmaceutical company databases. However, of the trove of public protein-ligand crystal structures, the vast majority (~94%) with temperature annotations were determined at cryogenic (cryo) temperature (≤200 K), typically after the protein crystals were flash-cooled in liquid nitrogen. By contrast, only a small minority (~6%) were determined at elevated temperatures (>200 K; of these, mostly >277 K or 0°C). This statistic is unnerving in light of the fact that, for proteins, cryo crystallography distorts protein conformational ensembles (*Keedy et al., 2014*), whereas room-temperature (RT) crystallography reveals distinct protein conformational heterogeneity, including alternate conformations of side chains and backbone segments, that better aligns with solution data and is in some cases more relevant to biological function (*Fraser et al., 2009*; *Fraser et al., 2011*; *Fenwick et al., 2014*; *Keedy et al., 2015*).

In contrast to proteins, relatively little is known about how crystallographic temperature affects protein-ligand interactions. Past studies that focused on individual compounds or small sets of related/congeneric compounds have offered tantalizing hints, with RT resulting in shifted binding poses (*Maeki et al., 2020*; *Bradford et al., 2021*; *Gildea et al., 2021*; *Milano et al., 2022*), binding at a different site (*Fischer et al., 2015*), and even a change of crystal symmetry (*Gildea et al., 2021*). In general, RT crystallography of protein-ligand complexes is increasingly accessible, thanks to advances in methodology (*Fischer, 2021*) including serial crystallography (*Milano et al., 2022*), even with as few as 1000 images, at synchrotrons (*Weinert et al., 2017*) or X-ray free electron lasers (*Moreno-Chicano et al., 2019*). Other RT approaches are also emerging, including in situ crystallography (*Sanchez-Weatherby et al., 2019*; *Lieske et al., 2019*), in some cases using crystallization plates pre-coated with dry compounds (*Gelin et al., 2015*; *Teplitsky et al., 2015*), as well as microfluidics (*Maeki et al., 2020*; *Sui et al., 2021*).

Despite this promising foundation, a central question remains: how frequently, and in what ways, does temperature affect protein-ligand structural interactions? To our knowledge, this question has not yet been addressed using a sufficiently large library of chemically diverse ligands. This gap is a significant obstacle toward a thorough understanding of how ligand and protein conformational heterogeneity interplay with one another to control biologically important phenomena such as enzyme catalysis and allosteric regulation. It also limits the potential of structure-based drug design (SBDD), given that cryo temperature is reported to degrade the utility of crystal structures for computational docking and binding free energy calculations (*Bradford et al., 2021*).

An emerging high-throughput approach to identifying protein-ligand hits is crystallographic small-molecule fragment screening, in which hundreds to thousands of 'fragments'' of drug-like small molecules are subjected to high-throughput crystal soaking and structure determination with a protein of interest. For example, a recent crystallographic fragment screen of the SARS-CoV-2 coronavirus's main protease (Mᵖʳᵒ) (*Douangamath et al., 2020*) provided dozens of starting points for crowd-sourcing the design of potent new small-molecule inhibitor candidates (*Achdout et al., 2022*), complementing another crystallographic screen of Mᵖʳᵒ using repurposed drug molecules (*Günther et al., 2021*).

Previously, several authors of the current study performed a crystallographic fragment screen of the archetypal protein tyrosine phosphatase, PTP1B (also known as PTPN1) (*Keedy et al., 2018*), a highly validated therapeutic target for diabetes (*Elchebly et al., 1999*), cancer (*Krishnan et al., 2014*), and neurological disorders (*Krishnan et al., 2015*) that has also been deemed 'undruggable' (*Zhang, 2017*; *Mullard, 2018*). That fragment screen produced X-ray datasets for 1627 unique fragments, of which 110 were clearly resolved in electron density maps at 12 fragment-binding sites scattered across the surface of PTP1B. Of the top three fragment-binding 'hotspots', one was previously validated with a non-covalent allosteric small-molecule inhibitor (*Wiesmann et al., 2004*), and another was validated with a new covalent allosteric inhibitor inspired by the fragment hits (*Keedy et al.,*

*2018*), thus highlighting fragment screening as a valuable tool for discovering allosteric footholds in proteins (*Krojer et al., 2020*). Importantly, however, all previously reported crystallographic fragment screens, including those mentioned above, were conducted at cryo temperature.

To elucidate the role of temperature in dictating protein-ligand interactions, here we have explored the use of large-scale crystallographic small-molecule fragment screening at RT. Specifically, we have performed RT crystallographic fragment screens of PTP1B with many of the same fragments used for the previous cryo screen (https://elifesciences.org/articles/36307), thereby allowing direct inferences regarding the effects of temperature. Our work uses 143 unique, chemically diverse ligands which, to our knowledge, is several-fold (4–5×) more than any previous RT crystallographic study. Moreover, we have used two complementary strategies for RT data collection. The two screens were performed with different diffraction data collection approaches, at different times, and with distinct but partially overlapping sets of fragments – so together they ensure that our overall conclusions are robust.

In both RT fragment screens, we observe that fewer fragments bind, and on average more weakly (with lower occupancy). However, many fragments that bind at RT do so with a variety of temperature-dependent differences, including unique binding poses, changes in solvation, totally new binding sites, and even distinct protein allosteric conformational responses (*Cui et al., 2017*; *Choy et al., 2017*; *Keedy et al., 2018*; *Hjortness et al., 2018*; *Hongdusit et al., 2020*; *Torgeson et al., 2022*) to ligand binding. Serendipitously, we also identify a fragment that binds covalently to a key lysine in the allosteric 197 site (*Keedy et al., 2018*), providing an intriguing new foothold for further allosteric inhibitor development.

Overall, this work provides new insights into ligandability and allostery in the important therapeutic target enzyme PTP1B. More broadly, it highlights the limitations of relying solely on cryo crystallography and the relative advantages of RT crystallography for elucidating interactions between ligands and proteins, with implications for a wide range of applications including SBDD.

## Results

### Two crystallographic fragment screens at RT

This work centers on two RT crystallographic screens of PTP1B: single-crystal (hereafter abbreviated as '1-xtal') and in situ. For both these two new RT screens and the prior cryo screen (*Keedy et al., 2018*), the procedures were identical for crystallization and crystal soaking with small-molecule fragments. However, the procedures differed in their approaches to crystal harvesting and diffraction data collection. In the previous cryo screen, the fragment-soaked crystals were harvested by hand with nylon loops, cryo-cooled in liquid nitrogen, and subjected to X-ray diffraction under a traditional cryo gas stream. In the new 1-xtal RT screen, the fragment-soaked crystals were harvested by hand with nylon loops, enclosed in plastic capillaries to prevent dehydration, and subjected to X-ray diffraction at ambient temperature. In the new in situ RT screen, the unharvested fragment-soaked crystals, still in the mother liquor solution in the crystallization plates, were subjected directly to X-ray diffraction at ambient temperature. See Materials and methods for further details about the experimental procedures. As outlined below, the crystallographic data and hit rates were similar for both RT screens, suggesting that the alternative protocols did not significantly impact the overall results.

**Table 1.** X-ray datasets collected for both room-temperature crystallographic screens.
The total datasets tally for the in situ screen derives from a larger number of partial datasets or 'wedges' that were merged (see Materials and methods). The cryo-hit and cryo-non-hit categories are defined in Materials and methods. Datasets from crystals soaked with DMSO only are included in the total datasets tally, but not in the unique fragments tally.

|  | 1-xtal | In situ |
| --- | --- | --- |
| Total raw X-ray datasets collected | 269 | 111 |
| Processed datasets with unique fragments | 86 | 80 |
| Cryo-hit | 38 | 48 |
| Cryo-non-hit | 48 | 32 |
| DMSO (negative control) | 7 | 20 |

For each RT screen, we used small-molecule fragments that were used previously for the cryo temperature crystallographic screen (*Keedy et al., 2018*) (see Materials and methods). Fragments were chosen from two categories: (1) *cryo-hits* which bound to the protein in the previous cryo screen, and (2) *cryo-non-hits* which were soaked into crystals but did not bind in the previous cryo screen. The 1-xtal RT screen used 59 cryo-hits and 51 cryo-non-hits, whereas the in situ RT screen used 48 cryo-hits and 32 cryo-non-hits. The fragment sets for the two screens were partially overlapping and complementary, with 23 fragments in common, of which 20 were cryo-hits and 3 were cryo-non-hits.

The fragment-soaked and control dataset for both screens are totaled and categorized in *Table 1*. Unless noted otherwise, the unique fragment datasets plus DMSO datasets for each screen were used for all subsequent analyses. As the two screens had 23 fragments in common, there were a total of 86+80–23=143 unique fragments overall across both RT screens.

The data were high resolution for both RT screens (*Figure 1*): the average resolution was 2.30 Å for 1-xtal and 1.99 Å for in situ, as compared to 2.10 Å for the previous cryo screen (*Keedy et al., 2018*). The slightly lower resolution of the 1-xtal data may be due to some degree of radiation damage, which was largely avoided by the in situ strategy (see Materials and methods). As outlined below, the results of the two screens are broadly very similar, and indeed identical for several fragments used in both screens (*Figure 4—figure supplement 4*), suggesting that radiation damage with the 1-xtal data was not a major factor in dictating our overall results. Additionally, a visual inspection of all the 1-xtal RT hits featured in this paper did not show any signs of local radiation damage (see Materials and methods).

## Identifying fragment-binding hits

Using these high-resolution datasets, we identified low-occupancy protein fragment-binding events using the PanDDA algorithm (*Pearce et al., 2017a*) and manual inspection and modeling (see Materials and methods). For the fragments that bound to PTP1B at cryo (cryo-hits) (*Keedy et al., 2018*), we then examined how many bound to PTP1B at RT. The initial hit rates from the event maps automatically generated by PanDDA were surprisingly low: 12/38 (32%) for 1-xtal and 7/48 (15%) for in situ. Additionally, for cryo-non-hits, PanDDA revealed only two binding events; both were for the same fragment in the same dataset (vide infra).

To identify hits that may have been missed by the automated PanDDA event identification algorithm, we manually generated RT event maps with the cryo value for 1-BDC, a quantity within PanDDA that is directly related to ligand-binding occupancy (*Pearce et al., 2017a*) (see Materials and methods). With this approach, we found five new binding events: three for the 1-xtal datasets and two for in situ datasets. This brought the new totals to 15/38 (39%) for 1-xtal and 9/48 (19%) for in situ, still fairly low hit rates.

This observation prompted us to reexamine how the many partial datasets or 'wedges'' obtained from in situ crystallography are assembled into complete datasets for use in subsequent steps including map calculation and PanDDA modeling (see Materials and methods). Recently, a new software called cluster4x was unveiled for pre-clustering X-ray datasets in the space of differences in structure factor amplitudes and/or Cα positions (*Ginn, 2020*). When applied to our past cryo PTP1B screen (*Keedy et al., 2018*), cluster4x identified previously unrecognized binding events (*Ginn, 2020*). The RT datasets are more isomorphous than the past cryo datasets (*Figure 1—figure supplement 1*). Nevertheless, to enhance isomorphism for our in situ screen, here we used cluster4x to pre-cluster in situ wedges (*Figure 1—figure supplement 2*) before merging within three main clustersthat are partially overlapping but qualitatively distinct from each other. We then assembled sets of similar wedges into complete datasets (see Material and methods) for input to PanDDA. This pre-clustering protocol resulted in five additional hits that were not previously observed with the all-wedges datasets, bringing the total RT hit rate for cryo-hits up from 9/48 (19%) to 14/48 (29%) for in situ.

Given the final cryo-hit reproduction rates of 39% and 29% for the RT screens, we investigated whether temperature affected the binding occupancy, or percent of unit cells in the crystal with a fragment bound. As an accessible proxy for occupancy, we examined PanDDA 1-BDC values. Many fragments have lower occupancy at RT than at cryo (*Figure 2*). This trend holds for cryo-hits that bind to the cryo site at RT either with the same pose (blue points in *Figure 2*) or with a new pose (orange points in *Figure 2*) (see *Table 2*).

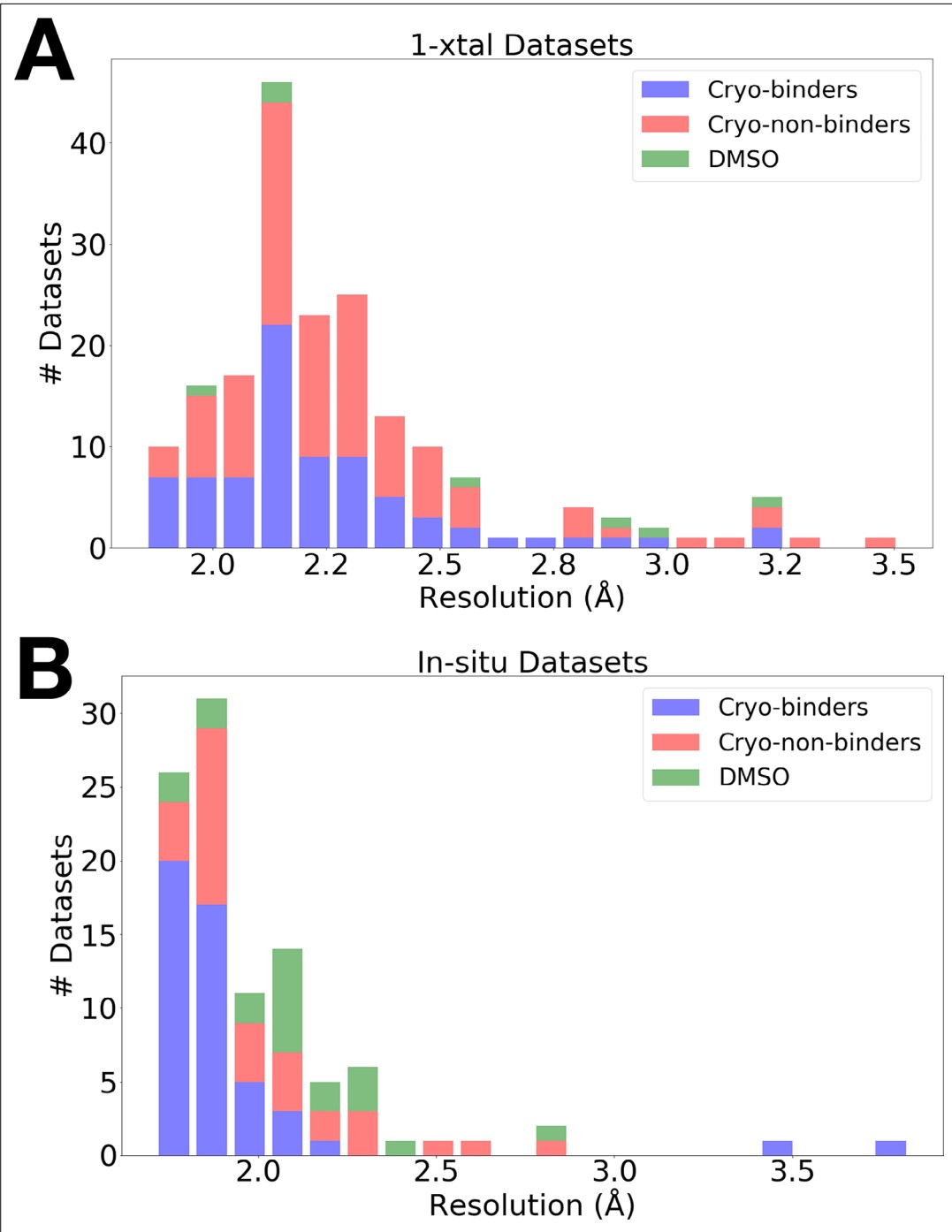

**Figure 1.** Resolution distributions from room-temperature (RT) crystallographic screens. Histogram of X-ray resolutions of datasets soaked with DMSO (green), cryo-hits compounds (blue), or cryo-non-hits (red), collected at RT via (**A**) 1-xtal or (**B**) in situ data collection techniques.

The online version of this article includes the following source data and figure supplement(s) for figure 1:

**Figure supplement 1.** Unit cell is more variable at cryogenic (cryo) than at room temperature (RT).

**Figure supplement 2.** Pre-clustering partial datasets from in situ crystallography.

**Figure supplement 2—source data 1.** Statistics for clusters of individual in situ wedges, prior to merging into complete datasets.

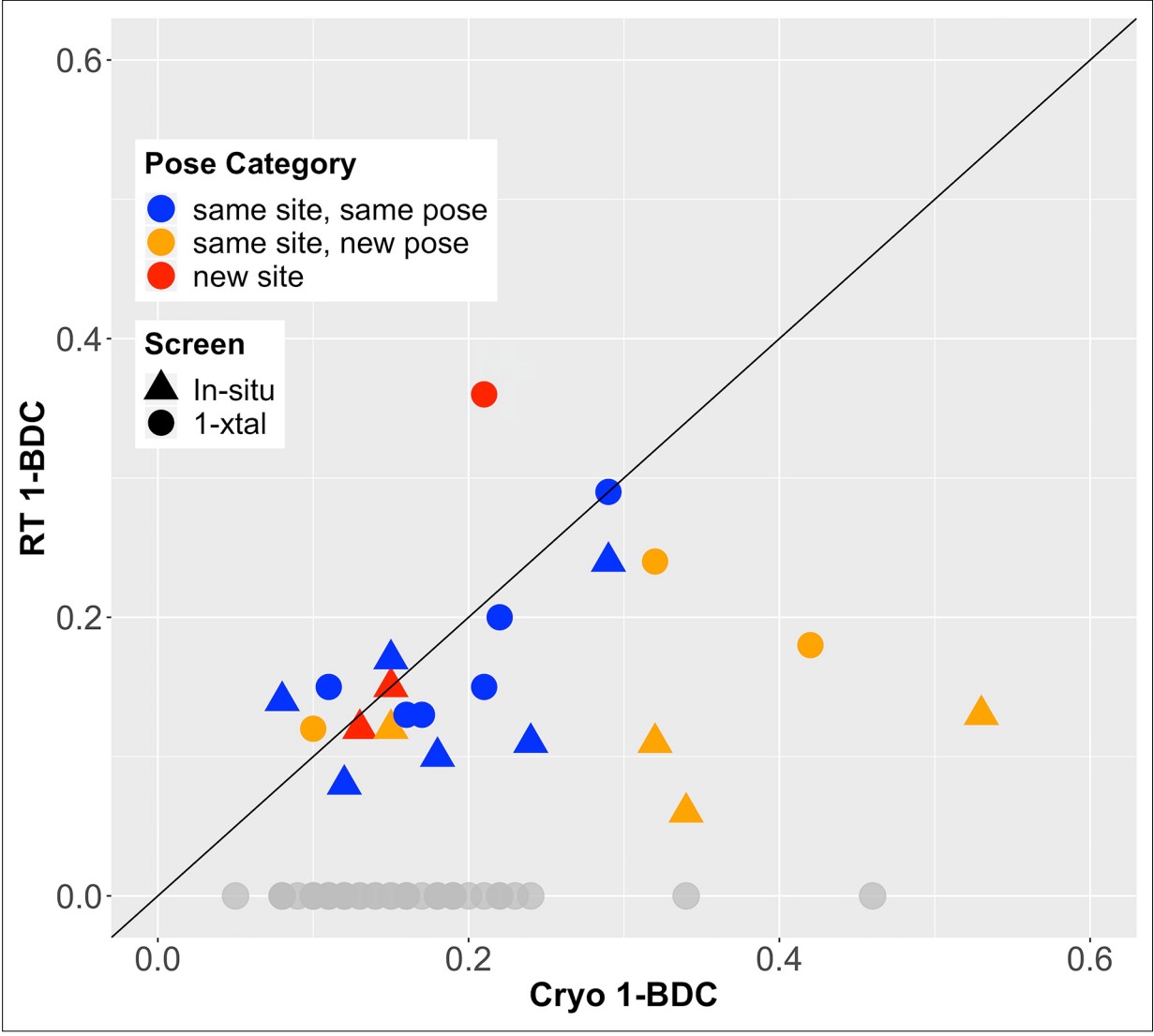

**Figure 2.** Fragment-binding occupancies are different and often lower at room temperature (RT). 1-BDC (a proxy for occupancy) is plotted for each binding event observed in either of two RT screens vs. in the previous cryogenic (cryo) screen. For two datasets, two binding events for the same fragment in the same structure are included as separate points. See *Table 2* for definitions of pose categories. Those that did not show binding at RT are in gray along the x-axis. In some additional cases, RT event maps were calculated using the cryo 1-BDC to identify bound ligands at RT; these cases would sit artificially on the diagonal, and are not shown here.

### Distribution of fragment hits at RT

Between our two RT screens, we have 15+14 = 29 new RT events with small-molecule fragments. These fragments fall into several categories based on the binding site, binding pose, and conformational response by the binding site (*Table 2*, *Table 2—source data 1*).

The RT fragment hits are bound at sites distributed throughout PTP1B, including at least one at the active site and at all three allosteric sites previously highlighted by the cryo screen (*Keedy et al., 2018*): the 197 site, BB site, and L16 site (*Figure 3—figure supplement 1*). Many of these sites have RT hits in both the 1-xtal and in situ screens (*Figure 3—figure supplement 1*), confirming the success of both screens. Notably, in all of these four key sites, one or more fragments bind differently from cryo – either binding with a new pose at RT or binding to this site only at RT and not cryo (*Figure 3*).

### Similar binding for many cryo-hits at RT

We next turned our attention to the precise binding poses of cryo-hit fragments at RT. Of the cryo-hits that also bind at RT, most do so with a similar pose (*Figure 4*, *Table 2*): 9 cases for 1-xtal

**Table 2.** Characteristics of fragment hits for room-temperature (RT) screens.

For each of the two RT screens, 1-xtal and in situ, all modelable fragment-binding events are categorized on the basis of the binding pose relative to the cryogenic (cryo) temperature binding pose. The tallies are on the basis of individual fragment-binding events, not crystal structures containing the fragment, because in some cases a given fragment binds at multiple sites, for example, both the cryo-hit site and a new site. Thus, the 'new site' category includes both cryo-hits that bind at a new site and cryo-non-hits that now bind at RT. The categories are not all mutually exclusive: for example, some fragments bind with the same site and same pose, but induce significant protein conformational change.

| RT pose category | 1-xtal | In situ | Total |
| --- | --- | --- | --- |
| Same site, same pose | 9 | 7 | 16 |
| Same site, new pose* | 4 | 5 | 9 |
| New site | 3 | 2 | 5 |
| Protein change | 2 | 3 | 5 |
| Not hit | 71 | 66 | 137 |

*The 'same site, new pose' category includes cases in which the fragment pose is the same but the protein conformation is notably altered at RT vs. cryo. See *Table 2—source data 1* for more information on which individual datasets fit into which categories.

The online version of this article includes the following source data for table 2:

**Source data 1.** List of all fragment hits for room-temperature (RT) screens.

**Source data 2.** Chemical properties of fragments and their binding sites.

(*Figure 4—figure supplement 1*) and 7 for in situ (*Figure 4—figure supplement 2*). Many of these are concentrated in two sites on the non-allosteric front side of the protein (*Figure 3—figure supplement 1*) that were also highly populated in the cryo screen. Additionally, some fragments are double-represented due to the overlap between the two screens. Notably, of the three fragments with binding events in both the 1-xtal and in situ screens, all three bind similarly in both RT screens (*Figure 4—figure supplement 4*), suggesting the RT results are reproducible and reliable.

Although all of the aforementioned fragments themselves bind with the same pose at RT vs. cryo, in some cases water molecules around them differ with temperature. In a few examples, clear event map density is present for a water at RT but not at cryo (*Figure 4C*, *Figure 4—figure supplement 5A–B*) or vice versa (*Figure 4—figure supplement 5C*), even when varying event map contour levels. Therefore, even when ligand binding is similar, the solvation layer around the ligand can change at cryo vs. RT.

## New binding poses at RT

Some cryo-hit fragments bind in the same site at RT, but with a quite different pose. In one striking example, the fragment binds with the central ring in the same position at RT vs. cryo, but with substantially different positions for the two substituent groups (*Figure 5*). The in-plane chlorine and out-of-plane methylamine group are clearly defined in the respective event maps: the RT density is incompatible with the cryo model, and vice versa. Notably, this fragment binds in the allosteric L16 site (*Figure 3*), which was first reported alongside the original cryo fragment screen for PTP1B and highlighted as a promising target for small-molecule allosteric inhibitor development (*Keedy et al., 2018*).

Another example features alternate ligand conformations that coexist in the same site, but only at one temperature. The RT event map suggests a pose with the carbonyl pointed one direction, toward Arg238 (*Figure 6A*). However, at cryo, this fragment was previously modeled with the carbonyl rotated by a 180° flip, enabling a water-bridged H-bond (*Figure 6B*). The RT event map has weak evidence at best for the flipped cryo conformation (*Figure 6A*). By contrast, the cryo event map has significant evidence for both conformations (*Figure 6B*). This observation is akin to other examples in which a ligand exhibits conformational heterogeneity in a single X-ray dataset (*van Zundert et al., 2018*). Here, however, the ligand conformational heterogeneity is temperature-dependent, enabling cross-pollination of conformations across temperatures to improve modeling (*Keedy, 2019*; *Bradford et al., 2021*).

A pair of other examples also feature fragments with distinct poses that are differentially stabilized at RT vs. cryo. In each of these two related examples, the RT event density is clear that the fragment binds with its longer substituent well ordered and pointed underneath the active-site WPD loop, which closes over the fragment (*Figure 7A and B*, left panels). At cryo, the loop still closes over the fragment, and the core of the fragment is in a similar location. However, the cryo event density is

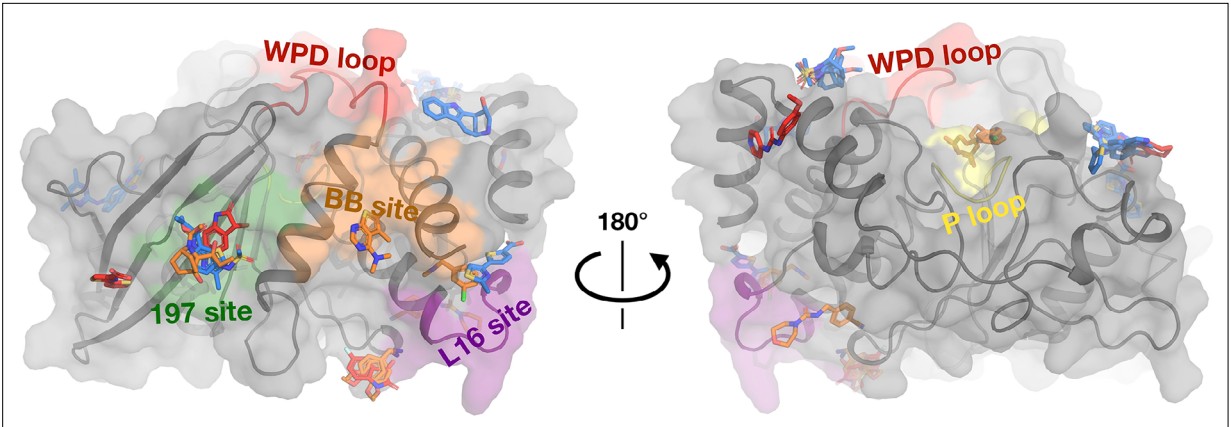

**Figure 3.** Fragments have a similar distribution across protein sites but different binding modes at room temperature (RT). Overview of fragments bound across PTP1B at RT, colored by RT pose compared to cryogenic (cryo) pose: same site, same pose (blue); same site, new pose (orange); new site (red). See *Table 2* for more details on the definitions of these classifications. Also highlighted are the active-site WPD loop (red), P loop (yellow), 197 allosteric site (green), BB allosteric site (orange), and L16 allosteric site (purple) (*Keedy et al., 2018*). The protein is shown in its open conformation with the WPD loop and L16 in the open state. The α7 helix is not shown since it is disordered when the protein is in the open state, which is favored at higher temperatures (*Keedy et al., 2018*). α7 does become ordered in one RT fragment-bound structure, but is not shown here.

The online version of this article includes the following figure supplement(s) for figure 3:

**Figure supplement 1.** Fragments bound at room temperature (RT) colored by RT screen.

inconsistent with the RT pose – instead, the longer substituent seems to protrude toward solution (*Figure 7A and B*, right panels). For one of these fragments, a new ordered water molecule at cryo displaces the RT ligand pose (*Figure 7B*).

## New binding sites at RT

Beyond just differences within the same binding site, temperature can also modulate ligand binding more dramatically, even altering what protein site the ligand binds to. In a first example, the fragment binds to the allosteric BB site (*Wiesmann et al., 2004*; *Keedy et al., 2018*) at cryo (*Figure 8A and C*), but there is no event density at RT. Instead, there is strong fragment-binding event density at a different site nearly 40 Å away (*Figure 8A and B*). The RT event density supports subtle protein shifts in the new binding site to accommodate the new fragment-binding event (*Figure 8B*). By contrast, in the cryo-binding site, the RT protein conformation would clash with the cryo pose, disallowing binding at RT.

In additional examples, elevated temperature dissipates what seem to be cryo-binding artifacts. In the first such example, at cryo the fragment binds with an artifactual stacking arrangement involving three copies of the fragment (*Figure 8D and E*), but at RT there is no event density (automated or custom) for this stacking. This result suggests that temperature can modulate protein-ligand energy landscapes significantly, in this case by disfavoring enthalpically favorable stacking at higher temperature. Moreover, at RT, new event density for a single copy of this fragment appears at a distal site (*Figure 8F*) that is over 45 Å away from the cryo site (*Figure 8D*). Cryo event density at the new site was too weak to justify modeling a bound fragment (*Keedy et al., 2018*). Thus, the cryo-binding site is unique to cryo and the RT-binding site is unique to RT. In fact, this is the only case in which a fragment binds at RT to a new site that was not previously thought to bind any fragments at cryo (although later computational reanalysis did discover one previously undetected adjacent cryo-hit in this area; *Ginn, 2020*). A Tris buffer molecule also fortuitously binds in the same location in another published structure (PDB ID 4y14), although it is held in place by a distinct crystal contact due to that structure's space group.

In a related but distinct case, a fragment previously bound at cryo with a seemingly similar artifactual stacking arrangement, this time involving two copies of the fragment (*Figure 8—figure supplement 1A*). However, at RT the entire stack does not disappear – instead, one copy remains bound (*Figure 8—figure supplement 1B*). At cryo, this latter copy was slightly more ordered than the other,

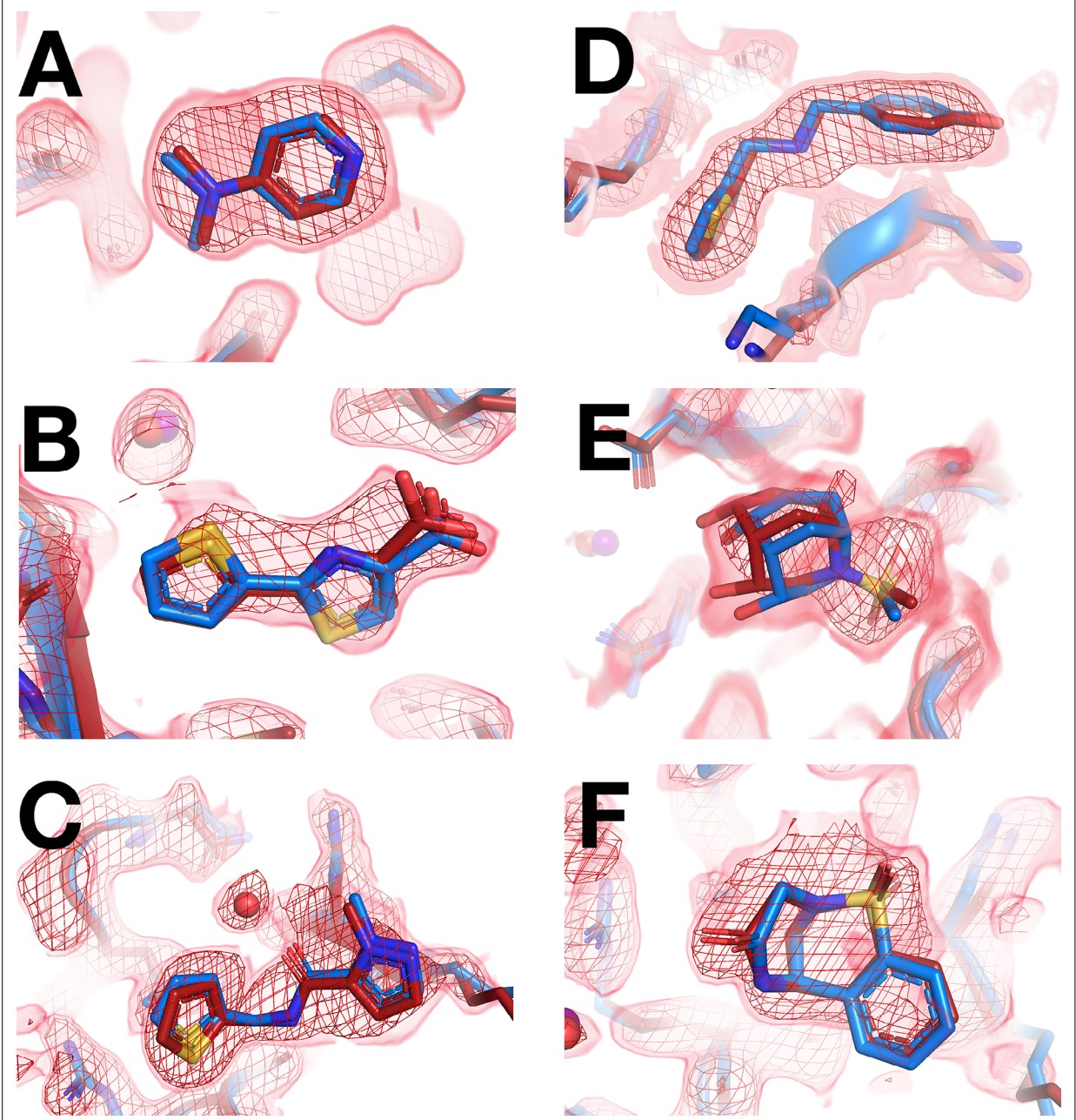

**Figure 4.** Fragments that bind similarly at room (RT) vs. cryogenic (cryo) temperatures. For each dataset, the RT PanDDA event map is in red (contour levels below), the RT model is in red (waters in red), and the corresponding cryo model is in blue (waters in purple). Datasets are named as follows: x####=RT in situ, z####=RT 1-xtal, y####=cryo. (A–C) in situ. (D–F) 1-xtal. (A) RT: x0224 (2.0 σ), cryo: y0118. (B) RT: x0285 (1.5 σ), cryo: y0772. (C) RT: x0262 (1.5 σ), cryo: y1656. (D) RT: z0007 (2.0 σ), cryo: y1710. (E) RT: z0015 (1.8 σ), cryo: y1554. (F) RT: z0025 (1.5 σ), cryo: y1294. This figure contains selected examples of fragments that bind similarly at RT vs. cryo; for all examples, see *Figure 4—figure supplement 1* for 1-xtal and *Figure 4—figure supplement 2* for in situ. For examples with no RT density for the cryo ligand using the cryo 1-BDC, see *Figure 4—figure supplement 3*.

The online version of this article includes the following figure supplement(s) for figure 4:

**Figure supplement 1.** All fragments that bind similarly at room (RT) vs. cryogenic (cryo) temperatures from 1-xtal screen.

**Figure supplement 2.** All fragments that bind similarly at room (RT) vs. cryogenic (cryo) temperatures from in situ screen.

**Figure supplement 3.** Fragments that bind at cryogenic (cryo) but not at room temperature (RT).

**Figure supplement 4.** Selected matching ligands across both screens have consistent binding poses.

**Figure supplement 5.** Differences in solvation around fragments at room (RT) vs. cryogenic (cryo) temperature.

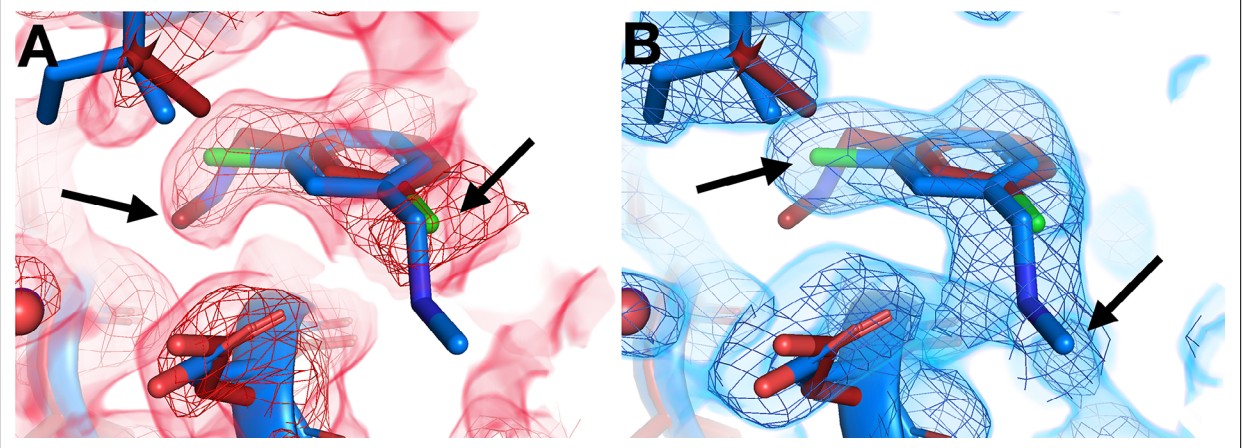

**Figure 5.** Temperature-dependent ligand conformational heterogeneity. (**A**) In the room temperature (RT) dataset (x0227), the RT event map contoured at 2 σ (red) matches the RT model (red) rather than the cryogenic (cryo) model (blue) for both substituent groups of the ring. (**B**) In the cryo dataset (y0071), the cryo event map contoured at 1.2 σ (blue) matches the cryo model rather than the RT model.

based on event map strength. Thus, elevated temperature is sufficient to displace the more weakly bound copy, but not the more tightly bound one.

In a final, somewhat more complicated example, a fragment previously bound at three distal sites at cryo. At RT the fragment binds to only one of the cryo sites, in a nearly identical pose. In the other cryo sites, RT-binding events were not readily detected, either automatically by PanDDA or in RT event maps calculated at the cryo events' 1-BDC values. More strikingly, at RT the fragment now binds to an additional new site (*Figure 8—figure supplement 2A and B*) that is over 40 Å away from any of the three cryo sites (*Figure 8—figure supplement 2C*). Although fragment binding was clear in cryo event maps at the three cryo sites, cryo density was unconvincing at the RT site; therefore, no binding event was detectable at this new site at cryo. Thus, as with the examples above (*Figure 8*), this fragment binds uniquely to a new site at RT.

## New covalent binding events to lysines

In addition to the fragments that switch binding sites at RT as detailed above, one fragment binds only in our RT datasets – and in an unexpected fashion. In RT event maps, we observe strong event

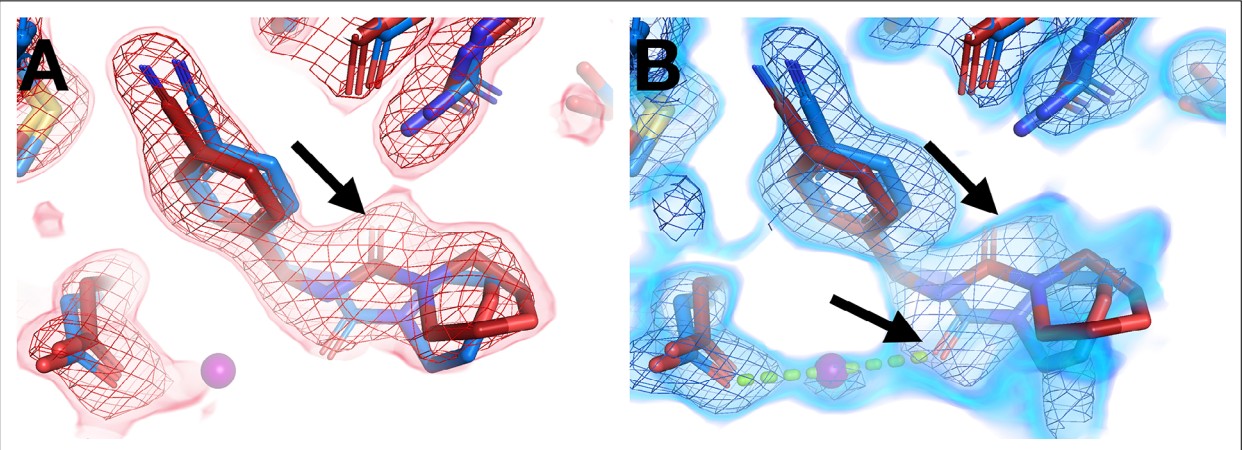

**Figure 6.** Temperature modulates fragment pose and solvation within the same site. (**A**) In the room temperature (RT) dataset (x0260), the RT event map contoured at 1.6 σ (red) matches the RT model (red), but shows little evidence for the cryogenic (cryo) model (y0180, blue). (**B**) In the corresponding cryo dataset, the cryo event map contoured at 1.6 σ (blue) matches both the cryo model (blue) and the RT model. Notably, only at cryo does the event map include density for a water molecule (purple ball) next to the fragment carbonyl group and well positioned for a hydrogen bond (pale green dashed line) with the cryo fragment pose.

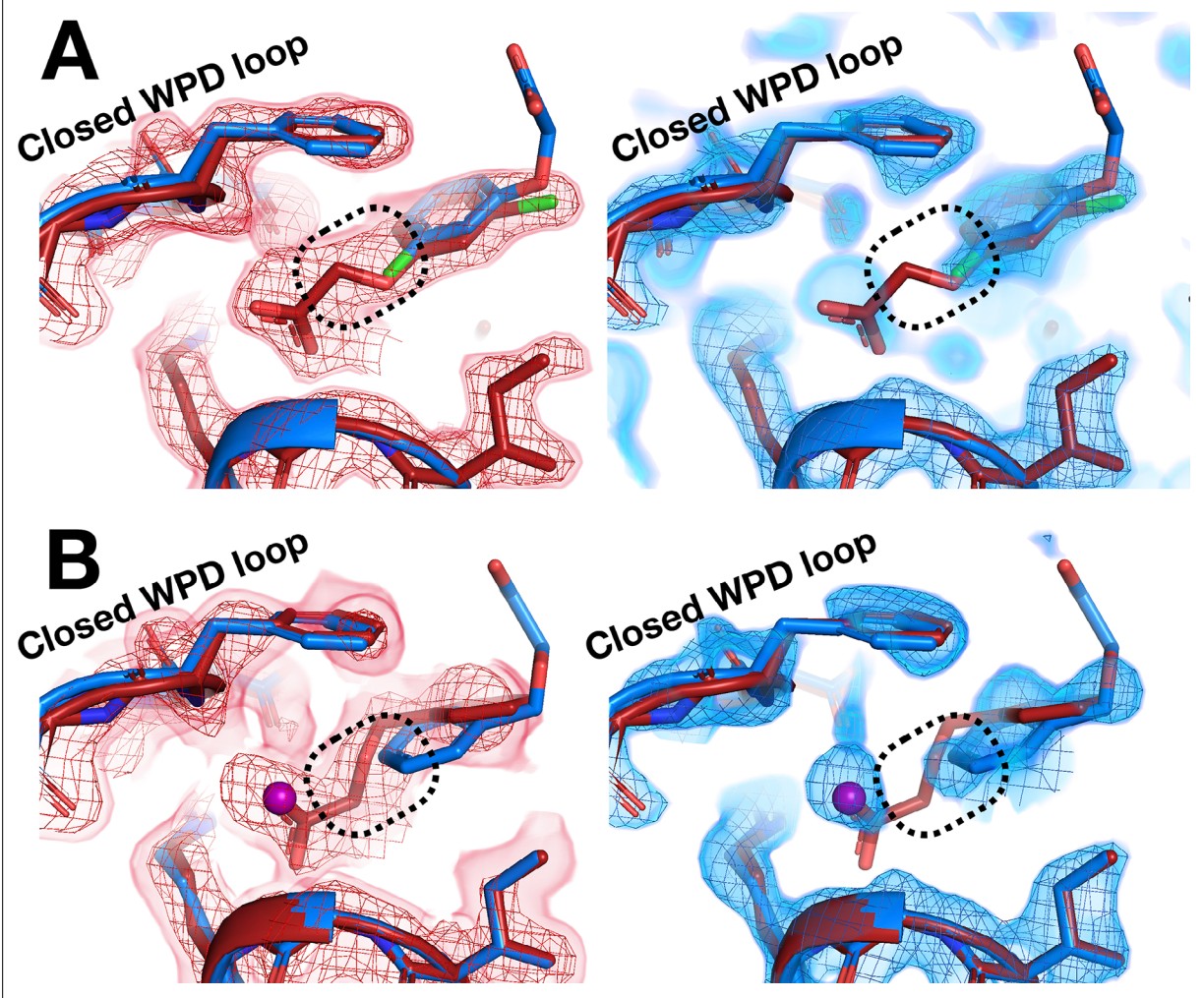

**Figure 7.** Room-temperature (RT) fragment pose is flipped compared to the cryogenic (cryo) pose. (**A**) *Left:* RT density (red) 1.5 σ, z0055 (red); y0884 (blue). *Right:* cryo density (blue) 1 σ, z0055 (red); y0884 (blue). (**B**) *Left:* RT density (red) 2 σ, x0256 (red); y0650 (blue). *Right:* cryo density (blue) 0.8 σ, x0256 (red); y0650 (blue) (not previously deposited to the PDB). Density is linked at RT (dashed box), consistent with the fragment pose, but is cut off at cryo, even at lower contour. There is little to no density for the open state of the WPD loop (not shown).

density at/near both the allosteric 197 and L16 sites (*Keedy et al., 2018*). Surprisingly, at each site, the event density is contiguous with the side chains of a nearby lysine residue (*Figure 9*), consistent with covalent binding by the isatin-based fragment. First, at the allosteric L16 site, the fragment binds covalently to Lys237 (part of the eponymous Loop 16) – although it binds adjacent to the L16 pocket itself, nearer to the allosteric BB site (*Figure 9C*). Second, at the allosteric 197 site site, the fragment binds covalently to Lys197 with a pose that is strikingly similar to that of a covalent allosteric inhibitor tethered to a K197C mutant (*Keedy et al., 2018*; *Figure 9A*).

The distal active-site P loop and substrate-binding loop adopt new conformations that are similar to those observed when the catalytic Cys215 is oxidized (*van Montfort et al., 2003*), although it is unclear whether Cys215 is oxidized in our RT event map. These conformations were not observed with the K197C-tethered allosteric inhibitor (*Figure 9—figure supplement 1*).

This fragment is a cryo-non-hit, meaning it demonstrably did not bind at cryo despite a high-resolution cryo dataset (1.89 Å, y1159). Indeed, it is the only cryo-non-hit to bind in either RT screen. This cryo-non-hit was chemically dissimilar to all previous cryo-hits: the most similar cryo-hit has a low Tanimoto score relative to this RT fragment (0.36, y1703) and does not bind near the RT sites. It is possible that the crystal for the cryo dataset was insufficiently soaked with this compound, or that the new RT-binding events seen here are due to additional chemical changes to the compound in DMSO solvent over time that altered its reactivity toward lysines. As expected for fragments due to their

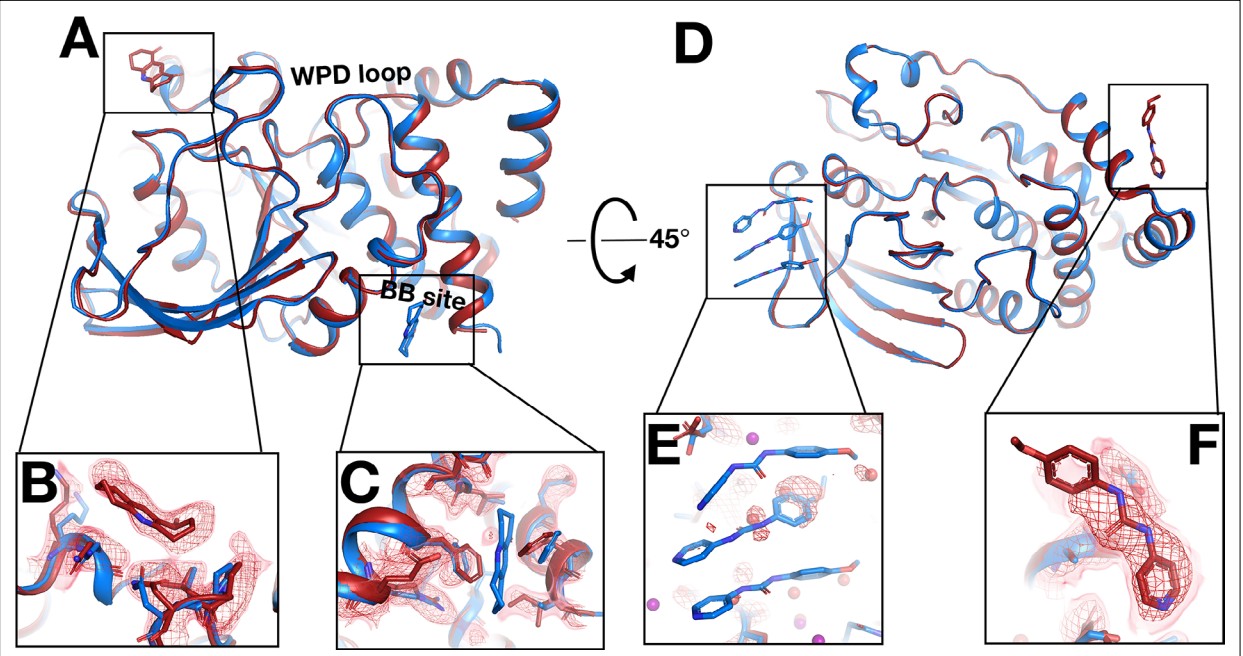

**Figure 8.** Fragments bind at new sites only at room temperature (RT). (**A–C**) First example. (**A**) The two sites are ~38 Å away from one another. (**B**) In the RT dataset (z0042), the RT event map, calculated with 1-BDC of 0.36 and contoured at 1.5 σ (red), supports a bound fragment in the RT model (red) at a new site while the cryogenic (cryo) model (y1525) (blue) has no bound fragment. (**C**) By contrast, the RT event map (same contour) does not show any density for the cryo model (blue) from the previous cryo dataset (y1525). (**D–F**) Second example. (**D**) The two sites are ~46 Å away from one another. (**E**) The RT event map contoured at 1.75 σ (red) (same contour) does not support the cryo model (blue) from the previous cryo dataset (y0572). (**F**) By contrast, at a new site the RT event map (same contour) supports a bound fragment in the RT model (x0225) (red). The cryo model has no bound fragment.

The online version of this article includes the following figure supplement(s) for figure 8:

**Figure supplement 1.** Only half of a cryogenic (cryo) stacking artifact disappears at room temperature (RT).

**Figure supplement 2.** Fragment that binds at a new site only at room temperature (RT).

weak binding affinities, this molecule does not inhibit PTP1B with an in vitro activity assay (*Keedy et al., 2018*) (data not shown). However, our observations here raise the hope that optimized versions of this compound, particularly driven by fragment linking of the K197C-targeted compound and this new fragment (*Figure 9A*), could yield potent allosteric inhibitors for wildtype (WT) PTP1B, without need for mutation to a cysteine.

## Unique protein conformational responses at RT

Temperature does not only affect fragment binding to the protein – it can also affect the protein's conformational response to fragment binding. With both screens, we observe protein conformational responses that are preferentially localized to the key allosteric sites that were identified in our previous study as being inherently linked to the active site (*Keedy et al., 2018*).

The C-terminal end of the α6 helix forms part of the allosteric L16 site (*Keedy et al., 2018*). At cryo, fragments in this site that intercalate below the α6 helix push it further in the direction of α7, the BB site, and the rest of the allosteric network (*Keedy et al., 2018*). At RT, structures with two of these fragments (*Figure 4—figure supplement 2G*, *Figure 5*) show that they affect the position of α6 similarly at RT vs. cryo (*Figure 10—video 1*); perhaps surprisingly, this remains true despite one fragment exhibiting a 180° pose flip (*Figure 5*).

However, in the nearby allosteric BB site (*Wiesmann et al., 2004*), the α6 helix is differentially ordered upon binding of a fragment at RT vs. cryo (*Figure 10*). Although the fragment binds in the same pose at RT and cryo, an entire additional helical turn of α6 is ordered at RT. This example illustrates that temperature can modulate not only the positions of protein structural elements during ligand binding, but also their relative order vs. disorder.

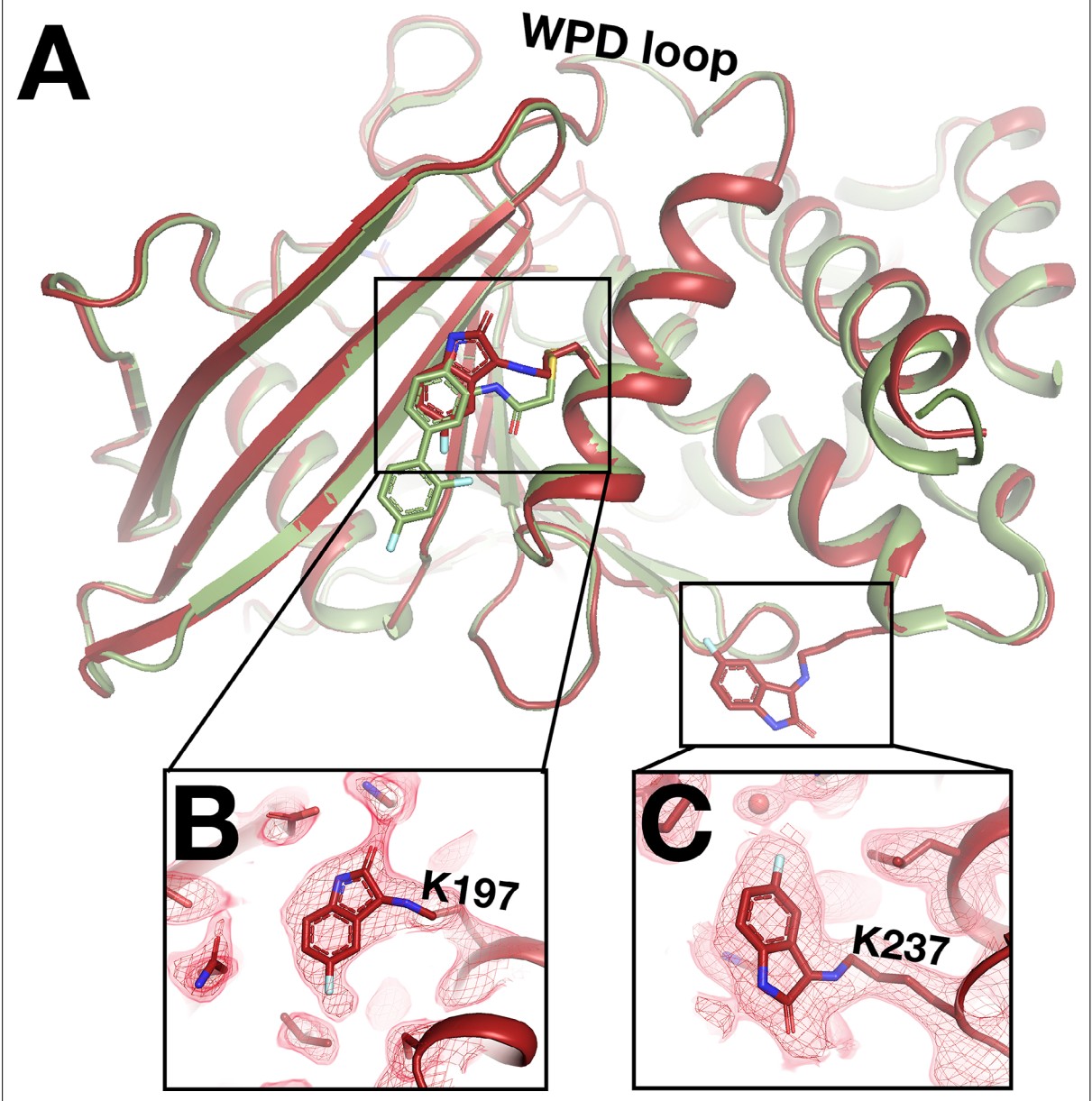

**Figure 9.** Unanticipated covalent adducts at previously reported allosteric sites only at room temperature (RT). (**A**) RT structure with the fragment covalently bound to both K197 and K237 (z0048, red), aligned with cryogenic (cryo) structure with a previously reported allosteric inhibitor covalently bound to K197C (6b95, green). (**B**) Fragment bound to K197 at the 197 allosteric site, with RT event density at 1.5 σ. (**C**) Fragment bound to K237 at the L16 allosteric site, with RT event density at 1.5 σ.

The online version of this article includes the following figure supplement(s) for figure 9:

**Figure supplement 1.** Conformational difference for the active-site P loop.

Elsewhere on the contiguous allosteric back face of PTP1B, in the 197 site (*Keedy et al., 2018*), a fragment binds with a similar pose at cryo and RT (*Figure 11*, *Figure 11—figure supplement 1*). When this fragment binds at cryo, the protein globally remains in its default open state (*Figure 11*). However, at RT, the allosteric L16 site closes, and the active-site WPD loop partially closes (*Figure 11*). Notably, this fragment binds in the same position as a previously reported covalently tethered allosteric inhibitor (*Keedy et al., 2018*; *Figure 11—figure supplement 2*; see also *Figure 9*). Thus, RT allows for distinct protein conformational redistributions in response to fragment binding in allosteric sites.

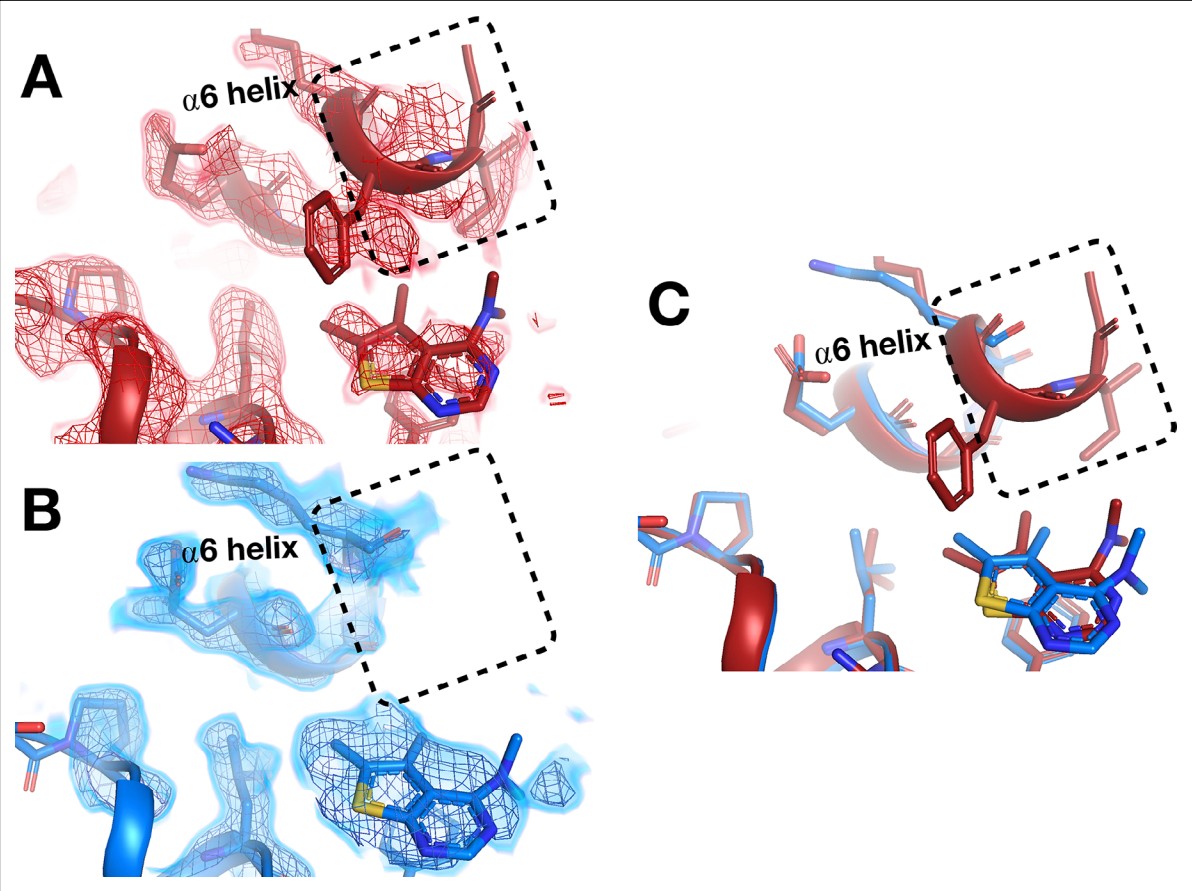

**Figure 10.** Temperature-dependent ordering of an α-helix augmenting a fragment-binding site. (**A**) In the BB allosteric site, the room temperature (RT) density, x0222 (red); contoured at 1.25 σ, is consistent with an extended and more ordered α6 helix (dashed box). (**B**) In contrast, the cryogenic (cryo) density, y0205 (blue); contoured at 1.75 σ, becomes disordered and therefore the α6 helix is not modeled as extended as in the RT model (dashed box). (**C**) Overlay of the two models showing the fragment pose is extremely similar whereas the RT helix is extended and more ordered (dashed box).

The online version of this article includes the following video for figure 10:

**Figure 10—video 1.** Fragments in allosteric L16 site shift the α6 helix to different extents.

https://elifesciences.org/articles/84632/figures#fig10video1

## Discussion

Cryo X-ray crystallography is the predominant experimental method for deriving insights into protein-ligand structures, but the effects of cryo temperature on protein-ligand binding are poorly understood. To fill this critical gap, here we report a large set of RT crystal structures of the dynamic enzyme PTP1B in complex with diverse small-molecule fragments, and present a detailed comparison with the corresponding cryo temperature structures. Our data suggest that temperature can significantly affect the occupancy, pose, and even location of small-molecule binding to proteins in crystal structures. Moreover, we show that temperature can modulate protein conformational responses to ligand binding, leading to new insights into allosteric networks.

Although only 29–39% of the fragments that previously bound at cryo temperature (*Keedy et al., 2018*) also bound here at RT, several lines of evidence suggest this is predominantly due to the difference in data collection temperature, as opposed to, for example, variability in experimental steps. First, the RT hit rates for cryo-hits were similar for our two RT screens, which were performed with different techniques (single-crystal and in situ) by largely different sets of people at different times. Second, we monitored log files from the acoustic droplet ejection instrument used for soaking (*Collins et al., 2017*) and excluded any crystals that may not have been soaked correctly. Third, in multiple RT datasets, a cryo-hit fragment demonstrably no longer binds at the original site but does bind at a different site (*Figure 8*), confirming the crystals were soaked correctly. Fourth, even when cryo-hit

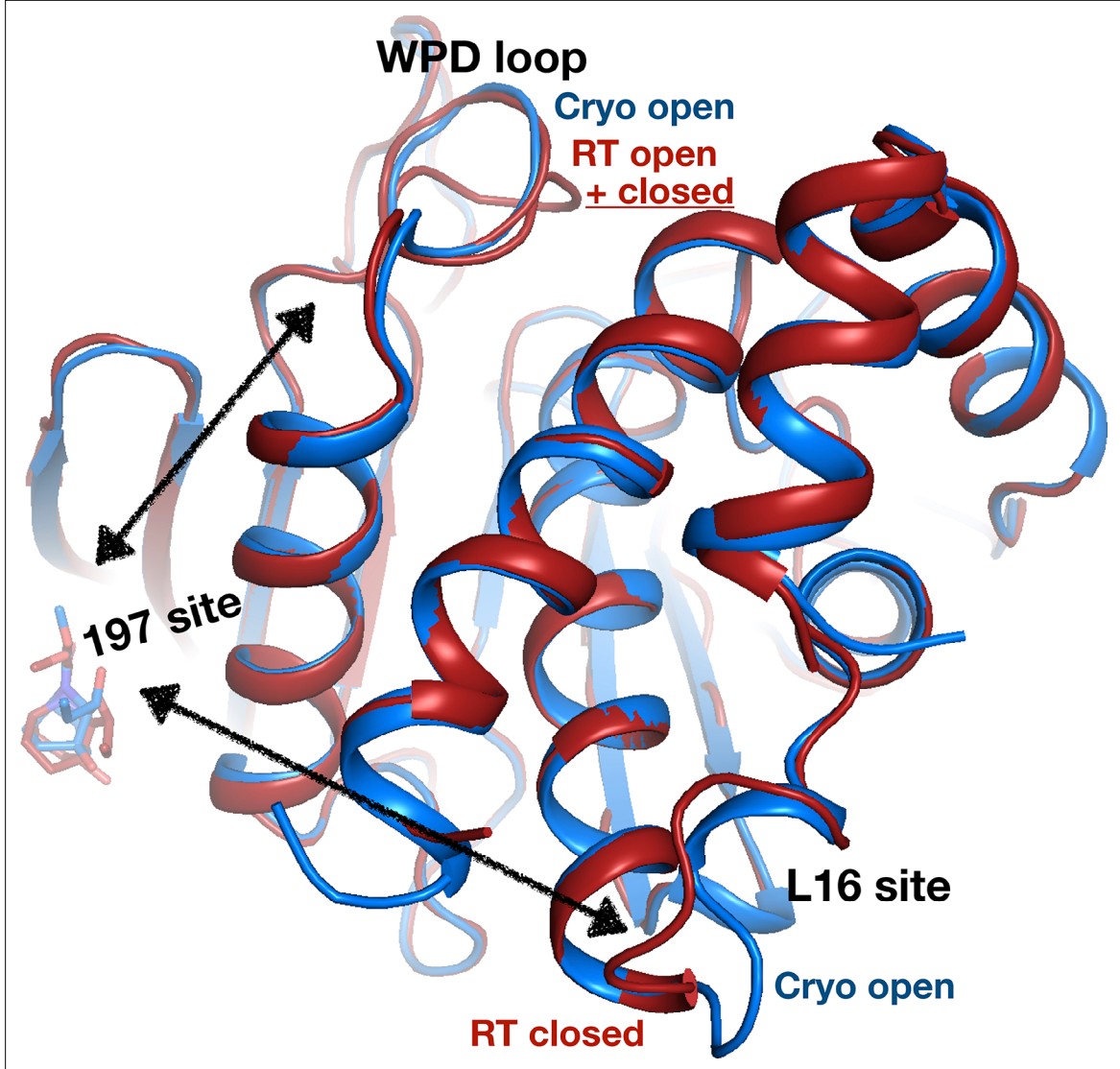

**Figure 11.** Allosteric protein responses at key sites seen only at room temperature (RT). Although the fragment binds in a similar manner and in the same allosteric site (the 197 site) in both the RT model (z0032) (red) and the cryogenic (cryo) model (y1763) (blue), the protein response is different between the two temperatures. At cryo, the protein retains the default open conformation, with loop 16 in the L16 site open and the WPD loop also open. Alternatively, at RT, the L16 site is fully closed, while the WPD loop exhibits alternate conformations with the loop both open and closed. The α7 helix (not shown) remains disordered in both temperatures.

The online version of this article includes the following figure supplement(s) for figure 11:

**Figure supplement 1.** Electron density evidence for allosteric protein responses at key sites seen only at room temperature (RT).

**Figure supplement 2.** Covalent allosteric inhibitor matches fragment with allosteric response at room temperature (RT).

fragments are observed in RT electron density event maps, we observe a trend of lower occupancies at RT (*Figure 2*). We conclude that the large temperature difference between cryo and RT (>178 K) underlies the observed changes in binding. This is in accord with recent studies in which only 0 of 9 (*Guven et al., 2021*) and 5 of 30 (*Gildea et al., 2021*) cryo-hit ligands were seen to bind at RT, and in which lower occupancies were seen at RT than at cryo for <10 ligands (*Bradford et al., 2021*).

When fragments do bind at RT, they often do so differently than at cryo, in a variety of ways (*Figure 4—figure supplement 5*, *Figure 5*, *Figure 6*, *Figure 8*, *Figure 8—figure supplement 2*). How does higher diffraction temperature cause such significant changes in protein-ligand binding? We speculate that after crystal soaking (which occurs at ambient temperature), all cryo-hit fragments are initially bound, but in many cases only loosely, with high B-factors that render them invisible

from RT diffraction data. During cryocooling, the ligand B-factors (i.e. temperature factors) then drop rapidly on a faster timescale than overall crystal cooling (*Halle, 2004*), with many becoming sufficiently well ordered to be observable in cryo density (at least with PanDDA). Relatedly, it is unclear why some fragments bind at both cryo and RT, but with a different pose or binding site: they have similar molecular weights and numbers of rotatable bonds, yet are more hydrophobic and have more interactions with the binding site (*Table 2—source data 2*). To more deeply interrogate the complex relationship between temperature, cooling kinetics, and protein-ligand conformational ensembles, additional experiments are planned using mechanically controlled variable cryocooling rate (*Warkentin et al., 2006*) and variable crystal size. Future studies can also explore the degree to which the conclusions drawn here for small-molecule fragments can be extrapolated to larger, drug-sized ligands.

Another relevant consideration is the expected variability of cryo structures, as a baseline for differences between RT vs. cryo structures. Previous work has shown that cryo crystal structures of proteins have greater inherent variability than do RT structures, presumably due to idiosyncratic crystal cryocooling kinetics (*Keedy et al., 2014*). However, despite growing interest in crystallographic fragment screening, no work has examined replicates of many fragment-soaked cryo crystal structures to establish the impact of crystal variability on details of fragment binding such as pose. One study using fragment screens with two different crystal forms of the same protein showed that most fragments did not bind in both crystal forms, and of those that did, only two of five bound in the same site with the same pose (*Schuller et al., 2021*); however, this is a different situation from repeats of the same fragment in the same crystal form. Another study showed that crystallographically refined occupancies of ligands approach saturation at ~15 min of soaking time (*Cole et al., 2014*) however, our soaking times were many hours (*Keedy et al., 2018*), so this should not be a significant source of variability in our datasets. The PanDDA algorithm seeks to overcome (typically cryo) dataset variability by averaging to establish a reliable ground state density estimate for the purposes of identifying hits, yet individual hits may still have idiosyncratic features. Overall, future studies focused on fragment (and larger ligand) reproducibility in terms of binding occupancy, site, and pose at cryo temperature would be useful contributions to the field.

Even when ligand binding is similar at RT vs. cryo, the protein response can differ. One case involves essentially complete closing of the allosteric L16 site, but only partial closing (~50%) of the active-site WPD loop (*Figure 11*) – in contrast to the previous paradigm in which the WPD loop and allosteric sites are precisely conformationally coupled (*Keedy et al., 2018*). Similar (de)coupling was also seen recently with serial synchrotron crystallography of apo PTP1B (*Sharma et al., 2023*). Thus, RT crystallography can add important nuance to our understanding of allosteric mechanisms in PTPs (*Choy et al., 2017*; *Cui et al., 2017*; *Hjortness et al., 2018*) and likely other proteins.

Our results here provide several insights that can aid future development of allosteric small-molecule modulators for PTP1B, a highly validated but 'undruggable' (*Zhang, 2017*; *Mullard, 2018*) therapeutic target. First, we observe new conformations for fragments on both sides of Loop 16 of the allosteric L16 site (*Figure 5*, *Figure 6*), offering unique footholds for structure-based inhibitor design of allosteric inhibitors. This local ligand heterogeneity, combined with the malleability of the adjacent α6 helix (*Figure 10*, *Figure 10—video 1*) and varying levels of apparent coupling between the L16 and active sites (*Figure 11*), argue for additional studies to decipher how different ligands in this region may selectively perturb the conformations of remote sites to allosterically control PTP1B function.

Second, one new RT fragment-binding site reported here was not previously shown to bind any fragments at cryo (*Keedy et al., 2018*) although additional clustering did identify one adjacent cryo-hit (*Ginn, 2020*), thus offering a new ligand-binding foothold. Coincidentally, the corresponding site in the paralog SHP2 has been successfully targeted with small-molecule allosteric inhibitors that stabilize a regulatory domain interface in the auto-inhibited state (*Chen et al., 2016*; *LaRochelle et al., 2018*). Although PTP1B lacks this additional regulatory domain, our data suggest future studies to explore whether it may nonetheless harbor latent allosteric capabilities that stem from this region within the catalytic domain.

Third, we observe a fragment covalently bound to Lys197 of the allosteric 197 site, with a similar pose as our previously reported allosteric inhibitor that was covalently tethered to an engineered K197C mutant (*Keedy et al., 2018*; *Figure 9*). This unexpected result opens new doors to design a covalent allosteric inhibitor targeting WT PTP1B, inspired by other success stories of progressing

covalent fragment hits (*Miller et al., 2013*; *Resnick et al., 2019*). The potential of targeting the allosteric 197 site of PTP1B is further reinforced by our new finding that fragment binding in this site (*Figure 11—figure supplement 1*) can elicit allosteric conformational responses at RT that were masked at cryo (*Figure 11*).

Altogether, we observe RT fragments bound in a variety of sites in PTP1B with potential for enabling downstream allosteric drug design. We see fragments bound in all three previously reported surface allosteric sites in the PTP1B catalytic domain: the BB site (*Wiesmann et al., 2004*), the 197 site (*Keedy et al., 2018*), and the L16 site (*Keedy et al., 2018*). The BB site is also thought to be near a secondary binding site for a second class of allosteric inhibitors for PTP1B, four example, MSI-1436, which primarily targets a different site in the disordered C-terminus (*Krishnan et al., 2014*). In addition to these three surface allosteric sites in the catalytic domain, we also see fragments bound in the active-site pocket (*Pedersen et al., 2004*). Notably, in all four of these key sites, we observe fragments that either adopt different poses at RT vs. cryo, or were not previously bound in that site at all at cryo (*Figure 3*). Such novel ligand poses in sites that are known to harbor allosteric capability offer promising new routes for fragment-based drug design (*Krojer et al., 2020*). This could be done either by 'growing' existing inhibitors by attaching moieties similar to fragment poses, or by designing new inhibitors 'from scratch' by identifying compounds that combine the (new) poses of multiple fragments in a site (*Gahbauer et al., 2023*). Fragment poses for these designs could derive from previous cryo structures and/or our new RT structures; the merits of combining multiple such sources of poses remain to be explored. Fragment-based design strategies could be used to develop non-covalent allosteric modulators or, in the case of the 197 site as mentioned above, covalent allosteric modulators of the WT enzyme (*Figure 9*). In addition to the fragments at previously established binding sites in PTP1B, as noted above we also see a fragment bound at a new site at RT: the N-terminal α1'-α2' helical bundle, corresponding to an allosteric inhibitor binding site in SHP2 (*Chen et al., 2016*; *LaRochelle et al., 2018*). This site was not bound by any fragments in the previous cryo screen (*Keedy et al., 2018*), making this new fragment a potentially valuable starting point for exploring the possible allosteric capabilities of this relatively underexplored region of the PTP1B catalytic domain tertiary structure.

It is instructive to consider the results reported here in light of the growing (and exciting) trend toward leveraging artificial intelligence and machine learning to address central problems in structural biology and biophysics. Most famously, the AI/ML algorithm AlphaFold 2 (*Jumper et al., 2021*) (and to a lesser extent RoseTTAfold; *Baek et al., 2021*) made a quantum leap in protein structure prediction accuracy. More relevant to the work reported here, AI/ML is being used to great effect for SBDD and computational chemistry, including protein-ligand docking (*Corso et al., 2022*) and ligand design (*Wallach et al., 2015*). Importantly, all of these methods rely on training data in the form of experimental protein structures from the PDB, the vast majority of which are cryo temperature crystal structures. For structure prediction, this temperature distribution undoubtedly introduces bias into the predicted models, likely favoring well-packed states that preclude functionally required conformational heterogeneity. For drug design, it may favor protein-ligand interactions that overweight enthalpic considerations and underweight entropic ones, feature inaccurate solvation environments, or suggest artificially rigid proteins. The full implications of these biases remain to be clarified (*Bradford et al., 2021*). RT crystal structures of protein-ligand interactions have the potential to ameliorate or bypass the limitations of cryo structures for training AI/ML methods. The number of structures reported here is insufficient to explore such ideas; it also remains unclear how useful weakly binding fragments may be for training AI/ML methods aimed at larger compounds. Nevertheless, our findings that protein-ligand interactions often differ from how they appear in cryo crystal structures prompts important questions as the age of AI/ML continues to rapidly unfold.

Overall, our work highlights the value and accessibility of RT crystallographic ligand screening for providing unique insights into protein-ligand interactions, particularly for allosteric sites (*Krojer et al., 2020*). More broadly, by using temperature as a readily accessible experimental knob, this study speaks to the potential of a multitemperature crystallography strategy, including excursions to higher temperatures in the physiological regime (*Doukov et al., 2020*; *Otten et al., 2020*; *Ebrahim et al., 2022*), for elucidating fundamental connections between molecular structure, heterogeneity, and function (*Keedy, 2019*).

## Materials and methods

**Key resources table**

| Reagent type (species) or resource | Designation | Source or reference | Identifiers | Additional information |
|---|---|---|---|---|
| Peptide, recombinant protein | Human PTP1B recombinant protein | This paper | | Purified from *Escherichia coli* BL21 cells |
| Software, algorithm | PanDDA software | PanDDA (https://pandda.bitbucket.io/) | | Version 0.2.14 |

### Protein expression

All experiments used the same PTP1B construct as was used previously: residues 1–321, WT* (C32S/C92V double mutation), in the pET24b vector carrying a kanamycin resistance gene (*Keedy et al., 2018*). Expression and purification were also performed as previously described (*Keedy et al., 2018*). PTP1B was transformed into BL21 *Escherichia coli* competent cells. The cultures were grown overnight in a 5 mL LB media containing 35 mg/L (final) kanamycin at 37°C shaking continuously at 150 rpm. Next, this overnight culture was used to inoculate 1 L LB media containing 35 mg/L (final) kanamycin. This culture was grown until the optical density at 600 nm ($OD_{600}$) reached between 0.6 and 0.8. PTP1B expression was then immediately induced by adding IPTG to 100 µM (final) and incubating for about 18–20 hr at 18°C shaking continuously at 200–250 rpm. The culture was then pelleted by centrifugation, the supernatant discarded, and the cell pellets ('cellets') harvested and stored at –80°C for subsequent purification.

### Protein purification

On the day of purification, each cellet was retrieved from –80°C and thawed on ice in 45 mL of lysis buffer (100 mM MES pH 6.5, 1 mM EDTA, freshly added 1 mM DTT) and a dissolved Pierce Protease Inhibitor Tablet. The cells were resuspended using a tabletop vortex. The homogenous cell suspension was then subjected to sonication using a Branson Digital Sonifier, with the probe submerged halfway into the suspension for about 20 min (10 s on/off) with 50% amplitude. The lysed cells were then subjected to centrifugation at 4°C, and the supernatant was filtered using 0.22 µm syringe filters and loaded onto an SP FF 16/10 cation exchange column, pre-equilibrated in lysis buffer, in an ÄKTA Pure purification system (GE Healthcare Life Sciences). The protein was eluted as 5 mL fractions using a linear gradient of lysis buffer from 0 to 1 M NaCl. PTP1B eluted at approximately 200 mM NaCl per the UV detector and SDS-PAGE analysis. The PTP1B fractions were pooled together and concentrated to 3 mL volume, then applied to a Superdex 75 (GE Healthcare Life Sciences) size exclusion column pre-equilibrated in crystallization buffer (10 mM Tris pH 7.5, 0.2 mM EDTA, 25 mM NaCl, 3 mM freshly added DTT). PTP1B eluted as a single peak, with high purity per SDS-PAGE analysis. The purified PTP1B protein was then concentrated to 40 mg/mL and used for crystallization.

### Protein crystallization

The PTP1B crystallization conditions used here were similar to those described previously (*Keedy et al., 2018*). 40 mg/mL protein in crystallization buffer was mixed with well solution (0.1 M HEPES pH 7.5, 0.3 M magnesium acetate, 13.5% PEG 8000, 2% ethanol, and 1 mM beta-mercaptoethanol) and seed stock in a 135:135:30 nL protein:well:seed ratio. Glycerol was not included. Seed stocks were prepared using Hampton Seed Bead tools with previously grown crystals. Drops were set using a TTP Labtech Mosquito device in 96-well sitting-drop crystallization trays. For the single-crystal screen, both MiTeGen In-Situ-1 and MRC SwissCi trays were used. For the in situ crystallographic screen, MiTeGen In-Situ-1 trays were used. Crystals appeared within about 3 days, and grew to maximum size within about 1 week. Crystals grew to dimensions of approximately $100\times20\times20$ µm$^3$ up to approximately $500\times100\times100$ µm$^3$.

### Fragment selection

For the 1-xtal screen, we used fragments from the Maybridge 1000 fragment library (Maybridge Ro3 core set), the Edelris Keymical fragment library, and the Diamond Light Source in-house fragment library (DSPL) (*Cox et al., 2016*). For cryo-hits, we included 59 fragments that bound to several

different sites of interest at cryo. For cryo-non-hits, we included 51 fragments that spanned the range of highly similar to dissimilar as compared to the previous cryo-hits.

For the in situ screen, we used fragments from the DSi-Poised (DSiP) library, which is a new version of the DSPL that contains many of the same fragments. For cryo-hits, we included all cryo-hits that were available in the DSiP library, as well as 12 cryo-hits we had previously purchased, for a total of 48 molecules. For cryo-non-hits, we included the 50 fragments in the DSiP library that were most similar to any previous cryo-hit. For both screens, similarity between fragments was assessed based on Tanimoto scores calculated using *RDKit, 2022*, topological fingerprints.

Some fragments that were cryo-non-hits in our original cryo screen (*Keedy et al., 2018*) were subsequently identified as cryo-hits using the new cluster4x method for computational clustering method (*Ginn, 2020*). Here, for both screens, we considered such fragments to be cryo-hits. This corresponded to three fragments for 1-xtal and one fragment for in situ. However, no RT-binding events were seen for any of these newly identified cryo-hits.

## Crystal soaking

For each screen, crystals were soaked with small-molecule fragments using an Echo acoustic droplet ejection liquid handler and a database mapping individual fragments to individual crystals, as described (*Collins et al., 2017*). For the in situ screen, anywhere from one to five wells were soaked with a given fragment, depending on the number of crystals per well.

Two strategies were used to confirm that fragments were successfully soaked into the crystallization drops. First, for both screens, log files for the acoustic droplet ejection device were inspected, and any wells with suspicious entries or errors were excluded. Second, for the in situ screen, optical images of the drops after soaking were visually inspected, and any wells that did not clearly feature a second adjacent drop corresponding to the fragment in DMSO were excluded.

## X-ray diffraction

For the 1-xtal screen, harvested crystals on size-matched nylon loops were enclosed in plastic capillaries containing ~10 μL of well solution and sealed with vacuum grease, and these samples were mounted onto the goniometer at Diamond Light Source beamline i03. Most datasets were collected with 180° of rotation over 1800 images with 0.1° oscillations with 0.05 s exposures. Some datasets near the end of the data collection shift were lowered to collect only 120° of crystal rotation, as smaller crystals sometimes did not appear to survive the full 180° dose. The X-ray beam was attenuated to 4.5% transmission for a flux of ~4.5e11 ph/s with a 50×20 or 80×20 μm$^2$ beam profile at a wavelength of 0.97625 Å. Temperature was controlled at 278 K using an Oxford Cryostream (800 Series).

For the in situ screen, crystallization trays were mounted onto the goniometer at Diamond Light Source beamline i24 for diffraction data collection. Partial datasets (wedges) were collected with up to 36° of rotation over 360 images with 0.1° oscillations with 0.03 s exposures. For each fragment, anywhere from 2 to 24 (average: 7) wedges were collected. In some cases, wedges for the same fragment derived from different crystals in the same well; in other cases, wedges for the same fragment derived from crystals in different wells soaked with the same fragment. The X-ray beam was attenuated to 1.5% transmission for a flux of ~4.5e10 ph/s with a 50×50 μm$^2$ beam profile at a wavelength of 0.96874 Å. Temperature was controlled by pointing a cryostream set to 277 K at the in situ tray mounted on the goniometer. Temperature was confirmed to be ~22°C (~295 K) by a handheld thermometer held by the tray.

Translational/vector data collection was not used for either screen. Whereas cryo datasets were previously named y#### (y for 'cryo'), RT datasets here were named x#### for the in situ screen and z#### for the 1-xtal screen.

## X-ray data processing

For the 1-xtal screen, datasets were reduced using XDS (*Kabsch, 2010*). The frames that were used to process the datasets were manually chosen to exclude frames where the number of detected spots dipped below around 20, commonly due to the crystal rotating out of the beam, the crystal reaching the end of its lifetime due to radiation damage, or when the diffraction quality dropped as a result of the dimensions of the crystal. Multiple datasets were merged only if they derived from the same crystal. Resolution cutoffs were chosen to ensure the following statistics in the highest

resolution bin: an <I/σ(I)> of 1.0 or higher, a completeness of 90% or higher, and a CC1/2 of at least 50%. The resolutions of individual datasets were not held to be identical, and the cutoff for each dataset was chosen to be the point at which the reflections from the highest resolution bin made the statistics of that bin better, or kept the same for <I/σ(I)>, $CC_{1/2}$ and completeness. Datasets shared a common set of $R_{free}$ flags and a common reference dataset to ensure consistent data indexing due to the space group of the crystal form, P 31 2 1. The final datasets were reasonably high resolution (*Figure 1A*).

For the in situ screen, individual wedges were first reduced using Dials (*Winter et al., 2018*). All frames were included. Resolution cutoffs for individual wedges were chosen automatically by Dials (*Winter et al., 2018*). Next, multiple wedges for the same fragment, regardless of crystal or well, were merged using xia2.multiplex (*Gildea et al., 2021*). In some cases the unit cell (89.6, 89.6, 106.2, 90, 90, 120) and/or the space group (P 31 2 1) was flagged in the xia2.multiplex input. Additionally, for some datasets the final merging step had to be done separately with dials.merge. For DMSO, anywhere from two to six wedges were merged. DMSO wedges were usually grouped by crystallization well, but in some cases were combined across wells to improve statistics. The final datasets were high-resolution (*Figure 1B*).

To check for global radiation damage, we used RADDOSE-3D to calculate the predicted average diffraction weighted dose (ADWD) for each dataset (*Bury et al., 2018*). For the in situ screen, predicted ADWD was ~0.03–0.04 MGy, depending on estimated crystal size, using the up-to-36° wedges. For the 1-xtal screen, predicted ADWD was ~3.2–7.4 MGy, depending on estimated crystal size and beam size, using the full 180° datasets. Thus the in situ data are well below the estimated RT limit of ~0.4 MGy (*Fischer, 2021*). The 1-xtal data are above the quoted RT limit (yet below the cryo limit of ~20–30 MGy; *Owen et al., 2006*); this was ameliorated for individual datasets by cutting later frames with reduced average intensities, as noted above. Additionally, we inspected 2Fo-Fc electron density maps for the individual 1-xtal RT hits featured here and observed no signatures of local radiation damage such as decarboxylation of Asp/Glu side chains, whether near the cryo and/or RT fragment-binding site(s) or elsewhere in the protein.

For an alternative data processing pipeline for the in situ data, the cluster4x algorithm (*Ginn, 2020*) was used to pre-cluster in situ wedges before merging with xia2.multiplex. First, the P 31 2 1 indexing hand for approximately half the wedges was changed using the Pointless (*Evans, 2006*) utility from CCP4 (*Winn et al., 2011*) to achieve consistency. Then, the wedges were clustered in real space. The resulting three clusters were partially overlapping in this space, and datasets were visually/manually assigned to these clusters. For each cluster, xia2.multiplex was used as described above, and separate PanDDa runs were performed as described below.

For each dataset, we used the Dimple utility from CCP4 (*Winn et al., 2011*) for phasing and initial refinement. Dimple was run with molecular replacement (flag: -M0) for the first dataset only, and only with downstream refinement steps (flag: -M1) for all other datasets. Additional flags were included to obtain a consistent set of $R_{free}$ reflections (`--free-r-flags`, `--freecolumn` R-free-flags). For both screens, the same structural model was used for Dimple, based on a high-resolution DMSO-soaked in situ merged dataset. This model reflects the predominant global open state of PTP1B, with the α7 helix unmodeled and the C-terminus of the α6 helix modeled with partial occupancy (*Keedy et al., 2018*).

## PanDDA modeling and refinement

For both the 1-xtal and in situ screens, PanDDA (*Pearce et al., 2017a*) version 0.2.14 was used. The pandda.analyse command was used with the minimum build datasets set to 20.

In addition to the automatic PanDDA analysis, for each dataset for which PanDDA did not show an event, we did a manual BDC scan from 1-BDC values of 0–0.9 as well as generating custom maps at the 1-BDC value that corresponded to the cryo 1-BDC. We saw five events with this manual inspection that PanDDA missed at the corresponding cryo 1-BDC. We used the automatically generated event maps throughout the manuscript, unless otherwise noted that a manually calculated event map is used.

Fragments and associated protein changes were modeled using pandda.inspect in Coot. Waters were kept the same between the unbound and bound models, except where the PanDDA event map indicated a shift, deletion, or an addition of a new water. Ligand restraints files were calculated with

eLBOW (*Moriarty et al., 2009*). We aimed to keep the RT models similar to the cryo models except when the RT map argued otherwise, so that modeled differences were due to temperature.

For the in situ datasets in this manuscript, we report all hits derived from the all-wedges datasets, plus a small number of distinct hits from the pre-clustered datasets as noted where appropriate.

Because ligands are not fully occupied, to prepare for refinement we must use an ensemble of bound state plus unbound, that is, ground state for refinement (*Pearce et al., 2017b*). We generated such an ensemble model with pandda.export. We then added hydrogens with Phenix ReadySet! Restraints, both between multi-state occupancy groups and between local alternate locations, were generated using giant.make_restraint scripts from PanDDA 1.0.0. The argument 'MAKE HOUT Yes' was added to the Refmac restraint file to ensure the Hydrogens were preserved.

For refinement of fragment-bound ensemble models, the published protocol for post-PanDDA refinement for deposition (*Pearce et al., 2017b*) was used, including the giant.quick_refine scripts from PanDDA 1.0.0 and the program Refmac (*Murshudov et al., 2011*). For a few examples, the script was rerun if the ligand was refined to a total occupancy greater than 1. Additionally, some hydrogens refined to 0 occupancy so they were manually edited to match the remainder of its residue. Refined bound-state models were then re-extracted using giant.split_conformations.

In addition to fragment-bound models, a ground-state model was refined for each screen, using the model used for MR previously and the highest-resolution DMSO dataset per screen.

## Acknowledgements

TS was the recipient of a fellowship award from the US Department of Education Graduate Assistance in Areas of National Need (GAANN) Program in Molecular Biophysics and Biomaterials at The City College of New York (PA200A150068), and is supported by CUNY Graduate Center Dissertation Fellowship. JTB was supported by an NSF GRFP award. DAK is supported by NIH R35 GM133769. We thank James Fraser for guidance and discussions; the beamline staff for help operating the XChem fragment-screening pipeline at Diamond Light Source; George Meigs, James Holton, James Sandy, and Juan Sanchez-Weatherby for help with initial in situ trial experiments; Helen Ginn for helpful discussions about diffraction dataset clustering; Virgil Woods and Nathanael Singh for help with a PTP1B activity assay; and Zachary Hill, Neel Shah, and Jack Taunton for helpful discussions about covalent fragment binding.

## Additional information

### Funding

| Funder | Grant reference number | Author |
| --- | --- | --- |
| US Department of Education | GAANN PA200A150068 | Tamar Skaist Mehlman |
| City College of New York | | Tamar Skaist Mehlman |
| National Science Foundation | GRFP | Justin T Biel |
| National Institutes of Health | R35 GM133769 | Daniel A Keedy |

The funders had no role in study design, data collection and interpretation, or the decision to submit the work for publication.

### Author contributions

Tamar Skaist Mehlman, Daniel A Keedy, Conceptualization, Resources, Data curation, Formal analysis, Supervision, Funding acquisition, Investigation, Visualization, Methodology, Writing – original draft, Project administration, Writing – review and editing; Justin T Biel, Syeda Maryam Azeem, Conceptualization, Data curation, Formal analysis, Funding acquisition, Investigation, Visualization, Methodology, Writing – original draft, Writing – review and editing; Elliot R Nelson, Sakib Hossain, Conceptualization, Data curation, Formal analysis, Investigation, Methodology, Writing – review and editing; Louise

Dunnett, Neil G Paterson, Data curation, Supervision, Investigation, Methodology, Writing – review and editing; Alice Douangamath, Romain Talon, Data curation, Investigation, Methodology, Writing – review and editing; Danny Axford, Helen Orins, Data curation, Investigation, Methodology, Project administration, Writing – review and editing; Frank von Delft, Resources, Data curation, Supervision, Investigation, Methodology, Project administration, Writing – review and editing

### Author ORCIDs
Tamar Skaist Mehlman (iD) http://orcid.org/0000-0002-8896-1628
Justin T Biel (iD) http://orcid.org/0000-0002-0935-8362
Syeda Maryam Azeem (iD) http://orcid.org/0000-0002-7712-527X
Sakib Hossain (iD) http://orcid.org/0000-0002-2602-9245
Frank von Delft (iD) http://orcid.org/0000-0003-0378-0017
Daniel A Keedy (iD) http://orcid.org/0000-0002-9184-7586

### Decision letter and Author response
Decision letter https://doi.org/10.7554/eLife.84632.sa1
Author response https://doi.org/10.7554/eLife.84632.sa2

## Additional files

### Supplementary files
• MDAR checklist

### Data availability
Bound state-models, structure factors, PanDDA event maps, and traditional maps (2Fo-Fc and Fo-Fc) for all fragment-bound structures are available in the Protein Data Bank under the following PDB ID accession codes: 7FQM, 7FQN, 7FQO, 7FQP, 7FQQ, 7FQR, 7FQS, 7FQT, 7FQU, 7FQV, 7FQW, 7FQX, 7FQY, 7FQZ, 7FRF, 7FRG, 7FRH, 7FRI, 7FRJ, 7FRK, 7FRL, 7FRM, 7FRN, 7FRO, 7FRP, 7FRQ, 7FRR. For each screen, a ground-state (unbound) model is also available, along with structure factors for all datasets involved in the respective screen, under the following PDB ID accession codes: 7FRE (1-xtal), 7FRS (in-situ), 7FRT (in-situ, cluster 1), 7FRU (in-situ, cluster 2). In addition, we provide a Zenodo directory containing our full PanDDA run directories, bound-state models, event maps, identifying information for all fragments, and related details at https://doi.org/10.5281/zenodo.7255364.

The following datasets were generated:

| Author(s) | Year | Dataset title | Dataset URL | Database and Identifier |
|---|---|---|---|---|
| Mehlman T, Biel J, Azeem SM, Nelson ER, Orins H, Hossain S, Dunnett LE, Talon R, Axford D, von Delft F, Keedy DA, Paterson NG, Douangamath A | 2022 | PanDDA analysis group deposition -- Crystal structure of PTP1B in complex with FMOPL000619a | https://www.rcsb.org/structure/7FQM | RCSB Protein Data Bank, 7FQM |
| Mehlman T, Biel J, Azeem SM, Nelson ER, Hossain S, Dunnett LE, Paterson NG, Douangamath A, Talon R, Axford D, Orins H, von Delft F, Keedy DA | 2022 | PanDDA analysis group deposition -- Crystal structure of PTP1B in complex with FMOOA000497a | https://www.rcsb.org/structure/7FQN | RCSB Protein Data Bank, 7FQN |

*Continued*

| Author(s) | Year | Dataset title | Dataset URL | Database and Identifier |
|---|---|---|---|---|
| Mehlman T, Biel J, Azeem SM, Nelson ER, Hossain S, Dunnett LE, Paterson NG, Douangamath A, Talon R, Axford D, Orins H, von Delft F, Keedy DA | 2022 | PanDDA analysis group deposition -- Crystal structure of PTP1B in complex with FMOOA000523a | https://www.rcsb.org/structure/7FQO | RCSB Protein Data Bank, 7FQO |
| Mehlman T, Biel J, Azeem SM, Nelson ER, Hossain S, Dunnett LE, Paterson NG, Douangamath A, Talon R, Axford D, Orins H, von Delft F, Keedy DA | 2022 | PanDDA analysis group deposition -- Crystal structure of PTP1B in complex with FMOOA000505a | https://www.rcsb.org/structure/7FQP | RCSB Protein Data Bank, 7FQP |
| Mehlman T, Biel J, Azeem SM, Nelson ER, Hossain S, Dunnett LE, Paterson NG, Douangamath A, Talon R, Axford D, Orins H, von Delft F, Keedy DA | 2022 | PanDDA analysis group deposition -- Crystal structure of PTP1B in complex with FMOOA000611a | https://www.rcsb.org/structure/7FQQ | RCSB Protein Data Bank, 7FQQ |
| Mehlman T, Biel J, Azeem SM, Nelson ER, Hossain S, Dunnett LE, Paterson NG, Douangamath A, Talon R, Axford D, Orins H, von Delft F, Keedy DA | 2022 | PanDDA analysis group deposition -- Crystal structure of PTP1B in complex with FMOOA000666a | https://www.rcsb.org/structure/7FQR | RCSB Protein Data Bank, 7FQR |
| Mehlman T, Biel J, Azeem SM, Nelson ER, Hossain S, Dunnett LE, Paterson NG, Douangamath A, Talon R, Axford D, Orins H, von Delft F, Keedy DA | 2022 | PanDDA analysis group deposition -- Crystal structure of PTP1B in complex with FMOOA000555a | https://www.rcsb.org/structure/7FQS | RCSB Protein Data Bank, 7FQS |
| Mehlman T, Biel J, Azeem SM, Nelson ER, Hossain S, Dunnett LE, Paterson NG, Douangamath A, Talon R, Axford D, Orins H, von Delft F, Keedy DA | 2022 | PanDDA analysis group deposition -- Crystal structure of PTP1B in complex with FMOMB000293a | https://www.rcsb.org/structure/7FQT | RCSB Protein Data Bank, 7FQT |
| Mehlman T, Biel J, Azeem SM, Nelson ER, Hossain S, Dunnett LE, Paterson NG, Douangamath A, Talon R, Axford D, Orins H, von Delft F, Keedy DA | 2022 | PanDDA analysis group deposition -- Crystal structure of PTP1B in complex with FMSOA000470b | https://www.rcsb.org/structure/7FQU | RCSB Protein Data Bank, 7FQU |

*Continued on next page*

*Continued*

| Author(s) | Year | Dataset title | Dataset URL | Database and Identifier |
| --- | --- | --- | --- | --- |
| Mehlman T, Biel J, Azeem SM, Nelson ER, Hossain S, Dunnett LE, Paterson NG, Douangamath A, Talon R, Axford D, Orins H, von Delft F, Keedy DA | 2022 | PanDDA analysis group deposition -- Crystal structure of PTP1B in complex with XST00000847b | https://www.rcsb.org/structure/7FQV | RCSB Protein Data Bank, 7FQV |
| Mehlman T, Biel J, Azeem SM, Nelson ER, Hossain S, Dunnett LE, Paterson NG, Douangamath A, Talon R, Axford D, Orins H, von Delft F, Keedy DA | 2022 | PanDDA analysis group deposition -- Crystal structure of PTP1B in complex with FMOCR000171b | https://www.rcsb.org/structure/7FQW | RCSB Protein Data Bank, 7FQW |
| Mehlman T, Biel J, Azeem SM, Nelson ER, Hossain S, Dunnett LE, Paterson NG, Douangamath A, Talon R, Axford D, Orins H, von Delft F, Keedy DA | 2022 | PanDDA analysis group deposition -- Crystal structure of PTP1B in complex with FMOPL000601a | https://www.rcsb.org/structure/7FQX | RCSB Protein Data Bank, 7FQX |
| Mehlman T, Biel J, Azeem SM, Nelson ER, Hossain S, Dunnett LE, Paterson NG, Douangamath A, Talon R, Axford D, Orins H, von Delft F, Keedy DA | 2022 | PanDDA analysis group deposition -- Crystal structure of PTP1B in complex with FMOPL000278a | https://www.rcsb.org/structure/7FQY | RCSB Protein Data Bank, 7FQY |
| Mehlman T, Biel J, Azeem SM, Nelson ER, Hossain S, Dunnett LE, Paterson NG, Douangamath A, Talon R, Axford D, Orins H, von Delft F, Keedy DA | 2022 | PanDDA analysis group deposition -- Crystal structure of PTP1B in complex with FMOMB000203a | https://www.rcsb.org/structure/7FQZ | RCSB Protein Data Bank, 7FQZ |
| Mehlman T, Biel J, Azeem SM, Nelson ER, Hossain S, Dunnett LE, Paterson NG, Douangamath A, Talon R, Axford D, Orins H, von Delft F, Keedy DA | 2022 | PanDDA analysis group deposition -- Crystal structure of PTP1B in complex with FMOPL000089a | https://www.rcsb.org/structure/7FRF | RCSB Protein Data Bank, 7FRF |
| Mehlman T, Biel J, Azeem SM, Nelson ER, Hossain S, Dunnett LE, Paterson NG, Douangamath A, Talon R, Axford D, Orins H, von Delft F, Keedy DA | 2022 | PanDDA analysis group deposition -- Crystal structure of PTP1B in complex with Z31222641 | https://www.rcsb.org/structure/7FRG | RCSB Protein Data Bank, 7FRG |

*Continued*

| Author(s) | Year | Dataset title | Dataset URL | Database and Identifier |
|---|---|---|---|---|
| Mehlman T, Biel J, Azeem SM, Nelson ER, Hossain S, Dunnett LE, Paterson NG, Douangamath A, Talon R, Axford D, Orins H, von Delft F, Keedy DA | 2022 | PanDDA analysis group deposition -- Crystal structure of PTP1B in complex with Z2856434762 | https://www.rcsb.org/structure/7FRH | RCSB Protein Data Bank, 7FRH |
| Mehlman T, Biel J, Azeem SM, Nelson ER, Hossain S, Dunnett LE, Paterson NG, Douangamath A, Talon R, Axford D, Orins H, von Delft F, Keedy DA | 2022 | PanDDA analysis group deposition -- Crystal structure of PTP1B in complex with Z321318226 | https://www.rcsb.org/structure/7FRI | RCSB Protein Data Bank, 7FRI |
| Mehlman T, Biel J, Azeem SM, Nelson ER, Hossain S, Dunnett LE, Paterson NG, Douangamath A, Talon R, Axford D, Orins H, von Delft F, Keedy DA | 2022 | PanDDA analysis group deposition -- Crystal structure of PTP1B in complex with Z2856434770 | https://www.rcsb.org/structure/7FRJ | RCSB Protein Data Bank, 7FRJ |
| Mehlman T, Biel J, Azeem SM, Nelson ER, Hossain S, Dunnett LE, Paterson NG, Douangamath A, Talon R, Axford D, Orins H, von Delft F, Keedy DA | 2022 | PanDDA analysis group deposition -- Crystal structure of PTP1B in complex with Z30820160 | https://www.rcsb.org/structure/7FRK | RCSB Protein Data Bank, 7FRK |
| Mehlman T, Biel J, Azeem SM, Nelson ER, Hossain S, Dunnett LE, Paterson NG, Douangamath A, Talon R, Axford D, Orins H, von Delft F, Keedy DA | 2022 | PanDDA analysis group deposition -- Crystal structure of PTP1B in complex with Z2856434917 | https://www.rcsb.org/structure/7FRL | RCSB Protein Data Bank, 7FRL |
| Mehlman T, Biel J, Azeem SM, Nelson ER, Hossain S, Dunnett LE, Paterson NG, Douangamath A, Talon R, Axford D, Orins H, von Delft F, Keedy DA | 2022 | PanDDA analysis group deposition -- Crystal structure of PTP1B in complex with Z509756472 | https://www.rcsb.org/structure/7FRM | RCSB Protein Data Bank, 7FRM |
| Mehlman T, Biel J, Azeem SM, Nelson ER, Hossain S, Dunnett LE, Paterson NG, Douangamath A, Talon R, Axford D, Orins H, von Delft F, Keedy DA | 2022 | PanDDA analysis group deposition -- Crystal structure of PTP1B in complex with Z915492990 | https://www.rcsb.org/structure/7FRN | RCSB Protein Data Bank, 7FRN |

*Continued*

| Author(s) | Year | Dataset title | Dataset URL | Database and Identifier |
|---|---|---|---|---|
| Mehlman T, Biel J, Azeem SM, Nelson ER, Hossain S, Dunnett LE, Paterson NG, Douangamath A, Talon R, Axford D, Orins H, von Delft F, Keedy DA | 2022 | PanDDA analysis group deposition -- Crystal structure of PTP1B in complex with Z744754722 | https://www.rcsb.org/structure/7FRO | RCSB Protein Data Bank, 7FRO |
| Mehlman T, Biel J, Azeem SM, Nelson ER, Hossain S, Dunnett LE, Paterson NG, Douangamath A, Talon R, Axford D, Orins H, von Delft F, Keedy DA | 2022 | PanDDA analysis group deposition -- Crystal structure of PTP1B in complex with XST00000245b | https://www.rcsb.org/structure/7FRP | RCSB Protein Data Bank, 7FRP |
| Mehlman T, Biel J, Azeem SM, Nelson ER, Hossain S, Dunnett LE, Paterson NG, Douangamath A, Talon R, Axford D, Orins H, von Delft F, Keedy DA | 2022 | PanDDA analysis group deposition -- Crystal structure of PTP1B in complex with XST00000217b | https://www.rcsb.org/structure/7FRQ | RCSB Protein Data Bank, 7FRQ |
| Mehlman T, Biel J, Azeem SM, Nelson ER, Hossain D, Dunnett LE, Paterson NG, Douangamath A, Talon R, Axford D, Orins H, von Delft F, Keedy DA | 2022 | PanDDA analysis group deposition -- Crystal structure of PTP1B in complex with Z2856434906 | https://www.rcsb.org/structure/7FRR | RCSB Protein Data Bank, 7FRR |
| Mehlman T, Biel J, Azeem SM, Nelson ER, Hossain S, Dunnett LE, Paterson NG, Douangamath A, Talon R, Axford D, Orins H, von Delft F, Keedy DA | 2022 | PanDDA analysis group deposition -- Crystal structure of PTP1B after initial refinement with no ligand modeled | https://www.rcsb.org/structure/7FRE | RCSB Protein Data Bank, 7FRE |
| Mehlman T, Biel J, Azeem SM, Nelson ER, Hossain S, Dunnett LE, Paterson NG, Douangamath A, Talon R, Axford D, Orins H, von Delft F, Keedy DA | 2022 | PanDDA analysis group deposition of ground-state model of PTP1B | https://www.rcsb.org/structure/7FRS | RCSB Protein Data Bank, 7FRS |
| Mehlman T, Biel J, Azeem SM, Nelson ER, Hossain S, Dunnett LE, Paterson NG, Douangamath A, Talon R, Axford D, Orins H, von Delft F, Keedy DA | 2022 | PanDDA analysis group deposition of ground-state model of PTP1B, using pre-clustering, cluster 1 | https://www.rcsb.org/structure/7FRT | RCSB Protein Data Bank, 7FRT |

*Continued*

| Author(s) | Year | Dataset title | Dataset URL | Database and Identifier |
|---|---|---|---|---|
| Mehlman T, Biel J, Azeem SM, Nelson ER, Hossain S, Dunnett LE, Paterson NG, Douangamath A, Talon R, Axford D, Orins H, von Delft F, Keedy DA | 2022 | PanDDA analysis group deposition of ground-state model of PTP1B, using pre-clustering, cluster 2 | https://www.rcsb.org/structure/7FRU | RCSB Protein Data Bank, 7FRU |
| Skaist Mehlman T, Biel JT, Keedy DA | 2022 | PanDDA analysis of PTP1B re-screened against fragment libraries at RT | https://doi.org/10.5281/zenodo.7255364 | Zenodo, 10.5281/zenodo.7255364 |

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
