## [Editor Report]

Based on two room-temperature X-ray crystallographic screens of the phosphatase PTP1B against two sets of chemical fragments, and by comparing the results from a previous cryo screen, the authors report the important observation that, in addition to overlapping but non-identical sets of hits compared to the cryo screen, the room-temperature screens lead to significant differences in terms of binding sites and poses for some of the hits. The study provides compelling support for the use of room-temperature X-ray crystallography in early-stage drug discovery and highlights that temperature should be used as a parameter in efforts to extract additional insight from such analyses.

---

## [Decision Letter]

**Decision letter after peer review:**

Thank you for submitting your article "Room-temperature crystallography reveals altered binding of small-molecule fragments to PTP1B" for consideration by *eLife*. Your article has been reviewed by 2 peer reviewers, and the evaluation has been overseen by a Reviewing Editor and Amy Andreotti as the Senior Editor. The following individual involved in review of your submission has agreed to reveal their identity: Yibing Shan (Reviewer #2).

Essential revisions:

(1) Present a more concise summary of the differences between the two protocols in the current work, as well as a discussion of how they compare to the protocol used in the previously published cryo-temperature fragment screen.

(2) Include a discussion of the expected variability of cryo-temperature crystallography experiments.

(3) Include additional discussion regarding the locations of the identified binding sites in the context of PTP1B function, especially whether the locations of the newly identified fragment binding sites can be exploited in drug discovery.

*Reviewer #1 (Recommendations for the authors):*

To improve the clarity of the manuscript, please provide a concise summary of the differences between the two protocols called "1-xtal" and "in-situ" in a single location. Currently, this information is scattered throughout the methods, discussion, and Results sections.

It would be helpful to include a discussion of the expected variability of cryo-temperature crystallography experiments, such as those performed by different groups at different times, in order to provide a better baseline for the expected variation of binding poses for the RT fragments.

If possible, it would be valuable to provide additional evidence of binding for the fragments that showed significant differences in binding compared to the previous cryo experiments.

The introduction, results, and Discussion sections contain some repetitive information that could be shortened by only including each concept once.

Regarding the mention of the SHP2 binding site: In addition to the catalytic domain, the allosteric SHP2 binding site is connected to one SH2 domain and a loop connecting the two SH2 domains. Many important interactions of the ligands binding to this site form with these additional domains and help stabilize the protein in an auto-inhibited conformation.

*Reviewer #2 (Recommendations for the authors):*

This reviewer suggests the authors include some additional discussion for PTP1B aficionados regarding the locations of the identified binding sites in context of PTP1B function as a phosphatase, especially the locations of the newly identified fragment binding sites. Can the new binding sites potentially result in PTP1B inhibition and thus be exploited in drug discovery?

---

## [Author Response]

Essential revisions:(1) Present a more concise summary of the differences between the two protocols in the current work, as well as a discussion of how they compare to the protocol used in the previously published cryo-temperature fragment screen.

We agree this would be a helpful addition to our manuscript. To address this suggestion, we deleted the Discussion paragraph about the strengths and weaknesses of the two methods relative to serial approaches, deleted the text in the Introduction that introduces the two screens, and placed the following text at the start of the Results section in the subsection titled “Two crystallographic fragment screens at room temperature” to provide a concise summary in one location of the manuscript:

“This work centers on two room-temperature crystallographic screens of PTP1B: single-crystal (hereafter abbreviated as “1-xtal”) and in-situ. For both these two new RT screens as well as the prior cryo screen (Keedy et al. 2018), the procedures were identical for crystallization and crystal soaking with small-molecule fragments. However, the procedures differed in their approaches to crystal harvesting and diffraction data collection. In the previous cryo screen, the fragment-soaked crystals were harvested by hand with nylon loops, cryo-cooled in liquid nitrogen, and subjected to X-ray diffraction under a traditional cryogenic gas stream. In the new 1-xtal RT screen, the fragment-soaked crystals were harvested by hand with nylon loops, enclosed in plastic capillaries to prevent dehydration, and subjected to X-ray diffraction at ambient temperature. In the new in-situ RT screen, the unharvested fragment-soaked crystals, still in the mother liquor solution in the crystallization plates, were subjected directly to X-ray diffraction at ambient temperature. See Methods for further details about the experimental procedures. As outlined below, the crystallographic data and hit rates were similar for both RT screens, suggesting that the alternative protocols did not significantly impact the overall results.”

Note that we left all text regarding the two screens in the Methods section untouched, to ensure all the relevant technical details are available.

(2) Include a discussion of the expected variability of cryo-temperature crystallography experiments.

We agree that the variability of repeated cryo-temperature crystallography experiments is a relevant consideration when comparing cryo to RT structures. We have added the following text to the Discussion regarding this point:

“Another relevant consideration is the expected variability of cryo structures, as a baseline for differences between RT vs. cryo structures. Previous work has shown that cryo crystal structures of proteins have greater inherent variability than do RT structures, presumably due to idiosyncratic crystal cryocooling kinetics (Keedy et al. 2014). However, despite growing interest in crystallographic fragment screening, no work has examined replicates of many fragment-soaked cryo crystal structures to establish the impact of crystal variability on details of fragment binding such as pose. One study using fragment screens with two different crystal forms of the same protein showed that most fragments did not bind in both crystal forms, and of those that did, only 2 of 5 bound in the same site with the same pose (Schuller et al. 2021); however, this is a different situation from repeats of the same fragment in the same crystal form. Another study showed that crystallographically refined occupancies of ligands approach saturation at ~15 minutes of soaking time (Cole et al. 2014); however, our soaking times were many hours (Keedy et al. 2018), so this should not be a significant source of variability in our datasets. The PanDDA algorithm seeks to overcome (typically cryo) dataset variability by averaging to establish a reliable ground state density estimate for the purposes of identifying hits, yet individual hits may still have idiosyncratic features. Overall, future studies focused on fragment (and larger ligand) reproducibility in terms of binding occupancy, site, and pose at cryo temperature would be useful contributions to the field.”

(3) Include additional discussion regarding the locations of the identified binding sites in the context of PTP1B function, especially whether the locations of the newly identified fragment binding sites can be exploited in drug discovery.

We note that the binding sites in the PTP1B are mentioned in the Results subsection “Distribution of fragment hits at room temperature”, Figure 3, Figure 3 —figure supplement 1, and various other points of the manuscript. Nevertheless, we agree that further discussion of the sites of the fragment hits in PTP1B and how they relate to opportunities for drug development would be helpful to many readers. We have therefore added the following paragraph to the Discussion:

“Altogether, we observe RT fragments bound in a variety of sites in PTP1B with potential for enabling downstream allosteric drug design. We see fragments bound in all three previously reported surface allosteric sites in the PTP1B catalytic domain: the BB site (Wiesmann et al. 2004), the 197 site (Keedy et al. 2018), and the L16 site (Keedy et al. 2018). The BB site is also thought to be near a secondary binding site for a second class of allosteric inhibitors for PTP1B, e.g. MSI-1436, which primarily targets a different site in the disordered C-terminus (Krishnan et al. 2014). In addition to these three surface allosteric sites in the catalytic domain, we also see fragments bound in the active site pocket (Pedersen et al. 2004). Notably, in all four of these key sites, we observe fragments that either adopt different poses at RT vs. cryo, or were not previously bound in that site at all at cryo (Figure 3). Such novel ligand poses in sites that are known to harbor allosteric capability offer promising new routes for fragment-based drug design (Krojer, Fraser, and von Delft 2020). This could be done either by “growing” existing inhibitors by attaching moieties similar to fragment poses, or by designing new inhibitors “from scratch” by identifying compounds that combine the (new) poses of multiple fragments in a site (Gahbauer et al. 2023). Fragment poses for these designs could derive from previous cryo structures and/or our new RT structures; the merits of combining multiple such sources of poses remain to be explored. Fragment-based design strategies could be used to develop non-covalent allosteric modulators or, in the case of the 197 site as mentioned above, covalent allosteric modulators of the WT enzyme (Figure 9). In addition to the fragments at previously established binding sites in PTP1B, as noted above we also see a fragment bound at a new site at RT: the N-terminal α1’-α2’ helical bundle, corresponding to an allosteric inhibitor binding site in SHP2 (Chen et al. 2016; LaRochelle et al. 2018). This site was not bound by any fragments in the previous cryo screen (Keedy et al. 2018), making this new fragment a potentially valuable starting point for exploring the possible allosteric capabilities of this relatively underexplored region of the PTP1B catalytic domain tertiary structure.”

Reviewer #1 (Recommendations for the authors):To improve the clarity of the manuscript, please provide a concise summary of the differences between the two protocols called "1-xtal" and "in-situ" in a single location. Currently, this information is scattered throughout the methods, discussion, and Results sections.

As noted above under Essential Revisions, we have added new text (and removed other text) in response to this suggestion.

It would be helpful to include a discussion of the expected variability of cryo-temperature crystallography experiments, such as those performed by different groups at different times, in order to provide a better baseline for the expected variation of binding poses for the RT fragments.

As noted above under Essential Revisions, we have added new text in response to this suggestion.

If possible, it would be valuable to provide additional evidence of binding for the fragments that showed significant differences in binding compared to the previous cryo experiments.

For larger, more drug-like compounds, this type of experimental validation of binding would indeed be important. With fragment screening, by contrast, the typical paradigm is different: the structures of fragment hits serve as fodder to design optimized ligands, which are then subjected to rigorous biophysical analysis. A major reason for this paradigm is that fragments bind to proteins very weakly. This expectation is borne out by our data: the fragments’ crystallographic occupancies in our structures are low (on the order of 10-50%) despite the fragments being present in the crystal mother liquor at double-digit mM concentrations (see Methods), which implies binding constants (Kd) in the mM range. We obtained qualitatively similar results in this regard in both the prior cryo screen and these new RT screens. It is possible these low in-crystal affinities are lower bounds on solution affinities as in theory the fragments may only partially penetrate the crystal lattice (even with the soak times of many hours that we used). Nevertheless, our crystallographic observations are consistent with the general expectation that fragments often bind too weakly for confident detection by biophysical or biochemical assays. Moreover, in the previous cryo screen paper, biochemical tests of several of the fragments in the library showed no measurable effect in solution (Keedy et al., 2018). At any rate, individual fragment hits are not the end goal in and of themselves: instead, their structures collectively map the ligandable surface area of proteins, and provide hypotheses about how one might design optimized ligands that bind with (much) higher affinities and modulate protein function.

The introduction, results, and Discussion sections contain some repetitive information that could be shortened by only including each concept once.

This is a fair point -- we have now simplified the text in several places of the manuscript. For example, we deleted most of the third paragraph of the Discussion which recapped the types of changes seen at RT vs. cryo for fragment binding.

Regarding the mention of the SHP2 binding site: In addition to the catalytic domain, the allosteric SHP2 binding site is connected to one SH2 domain and a loop connecting the two SH2 domains. Many important interactions of the ligands binding to this site form with these additional domains and help stabilize the protein in an auto-inhibited conformation.

The reviewer is correct on this point. To better explain this context, we have added the following text to this paragraph:

“Second, one new RT fragment-binding site reported here was not previously shown to bind any fragments at cryo (Keedy et al. 2018) (although additional clustering did identify one adjacent cryo-hit Ginn 2020), thus offering a new ligand-binding foothold. Coincidentally, the corresponding site in the paralog SHP2 has been successfully targeted with small-molecule allosteric inhibitors that stabilize a regulatory domain interface in the auto-inhibited state (Chen et al. 2016; LaRochelle et al. 2018). Although PTP1B lacks this additional regulatory domain, our data suggest future studies to explore whether it may nonetheless harbor latent allosteric capabilities that stem from this region within the catalytic domain.”

Reviewer #2 (Recommendations for the authors):This reviewer suggests the authors include some additional discussion for PTP1B aficionados regarding the locations of the identified binding sites in context of PTP1B function as a phosphatase, especially the locations of the newly identified fragment binding sites. Can the new binding sites potentially result in PTP1B inhibition and thus be exploited in drug discovery?

As noted above under Essential Revisions, we have added new text in response to this suggestion.